# TEST-TIME MATCHING: UNLOCKING COMPOSITIONAL REASONING IN MULTIMODAL MODELS

**Yinglun Zhu**[†]  **Jiancheng Zhang**  **Fuzhi Tang**
University of California, Riverside
{yzhu, jzhan745, fuzhit}@ucr.edu

## ABSTRACT

Frontier AI models have achieved remarkable progress, yet recent studies suggest they struggle with *compositional reasoning*, often performing at or below random chance on established benchmarks. We revisit this problem and show that widely used evaluation metrics systematically *underestimate* model capability. To correct this artifact, we introduce a *group matching score* that more faithfully evaluates model capability. Moreover, correctness under the new metric can be translated into correctness under existing metrics via a simple overfitting step. This adjustment enables SigLIP-B16 to surpass all previous results and GPT-4.1 to *yield the first result surpassing estimated human performance on Winoground.* Building on this insight, we propose *Test-Time Matching* (TTM), an iterative, self-improving algorithm that further bootstraps model performance without any external supervision. TTM delivers additional, non-trivial improvements: for example, TTM **enables SigLIP-B16 to surpass GPT-4.1 on MMVP-VLM, establishing a new state of the art**. TTM also extends beyond contrastive vision-language models, yielding clear gains on a generative multimodal model across benchmarks. Importantly, TTM remains broadly effective even on benchmarks without metric-induced effects or group structures, **achieving relative gains up to 85.7%** on challenging datasets such as WhatsUp. Across 16 dataset variants spanning diverse setups, our experiments demonstrate that TTM consistently improves model performance and advances the frontier of compositional reasoning.

## 1 INTRODUCTION

Compositional reasoning provides a stringent test of frontier AI models, assessing their ability to systematically combine primitive elements—such as objects, attributes, and relations—to interpret or reason about novel configurations (Lake et al., 2017; Bahdanau et al., 2019). Recent benchmarks evaluate this capability by organizing examples into small groups of images and captions that differ in subtle yet systematic ways (Thrush et al., 2022; Hsieh et al., 2023; Kamath et al., 2023; Tong et al., 2024; Burapacheep et al., 2024). For example, Winoground consists of $2 \times 2$ groups where both captions contain the same words but in different orders, such that each caption correctly describes only one of the two images.

Despite the impressive practicality of modern multimodal systems, both contrastive vision-language models (VLMs) and multimodal large language models (MLLMs) have been reported to perform at or below random guessing on these benchmarks (Thrush et al., 2022; Diwan et al., 2022; Tong et al., 2024; Burapacheep et al., 2024; Li et al., 2025). On Winoground, even frontier AI models still fall far short of the estimated human performance of $85.5$ (Thrush et al., 2022), with the previous state of the art reaching only $58.75$, achieved through scaffolding and prompt tuning GPT-4V (Wu et al., 2023; Vaishnav & Tammet, 2025).

We revisit this conclusion and show that the widely used evaluation metric GroupScore (Thrush et al., 2022; Tong et al., 2024; Burapacheep et al., 2024) systematically *underestimates* model capability. We introduce a *group matching score* (GroupMatch) that more faithfully evaluates model capability. Importantly, correctness under GroupMatch can be translated into correctness under GroupScore

---

[†]Project lead and corresponding author. Code: https://github.com/yinglunz/test-time-matching

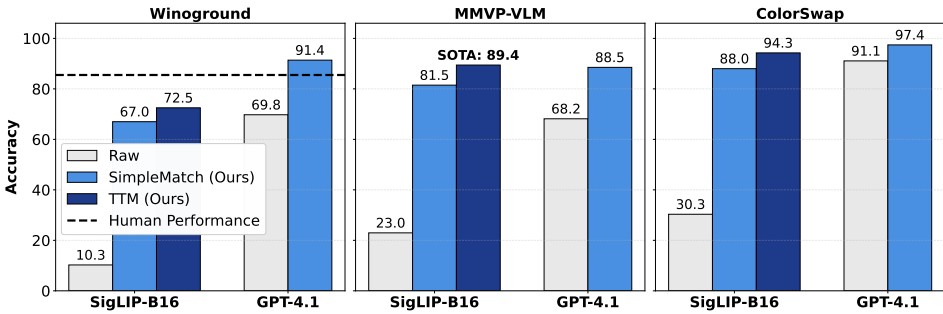

Figure 1: SimpleMatch and TTM substantially improve VLM and MLLM performance on compositional reasoning benchmarks Winoground, MMVP-VLM, and ColorSwap, achieving new performance records. We highlight: (1) SimpleMatch enables GPT-4.1 to surpass human performance on Winoground (*left*), and (2) TTM enables SigLIP-B16 to surpass GPT-4.1 on MMVP-VLM, establishing a new state of the art (*middle*).

by overfitting to the matchings induced by GroupMatch, an approach we refer to as SimpleMatch (see Section 3.1). SimpleMatch alone reveals substantial hidden capability: as shown in Fig. 1, SigLIP-B16 improves from $10.25 \rightarrow 67$ on Winoground, $22.96 \rightarrow 81.48$ on MMVP-VLM, and $30.33 \rightarrow 88$ on ColorSwap, surpassing all previous results without access to additional data (Wu et al., 2023; Vaishnav & Tammet, 2025; Zhang et al., 2024c; Burapacheep et al., 2024). GPT-4.1 also improves dramatically, from $69.75 \rightarrow 91.38$ on Winoground, $68.15 \rightarrow 88.52$ on MMVP-VLM, and $91.08 \rightarrow 97.42$ on ColorSwap—*yielding the first result to surpass the estimated human performance of 85.5 on Winoground* (Thrush et al., 2022).[1]

Building on this insight, we introduce *Test-Time Matching* (TTM), an iterative, self-improving algorithm that further bootstraps model performance without any external supervision. TTM selects matching-induced pseudo-labels for self-training and progressively relaxes the selection threshold to expand coverage over the test set. This yields *additional, non-trivial* gains on top of SimpleMatch: SigLIP-B16 reaches 72.5 on Winoground, 89.44 on MMVP-VLM, and 94.25 on ColorSwap. Remarkably, TTM elevates SigLIP-L16 to the level of GPT-4.1 on ColorSwap (Table 1) and **enables SigLIP-B16 to surpass GPT-4.1 on MMVP-VLM, establishing a new state of the art.** See Fig. 1 and Table 1 for details. TTM also extends beyond contrastive vision-language models, yielding clear gains on a generative multimodal model across benchmarks (Table 2). Crucially, TTM is broadly effective even where metric changes cannot help—on $1 \times k$ benchmarks such as SugarCrepe (Hsieh et al., 2023) and WhatsUp (Kamath et al., 2023), where GroupScore and GroupMatch coincide, TTM still delivers substantial test-time improvements, including **up to 85.7% relative gains** on challenging datasets such as WhatsUp (Fig. 3).

Finally, we extend TTM beyond group-structured datasets by formulating a single global matching across all images and captions. Even a one-shot global matching outperforms raw GroupScore, and applying the global variant of TTM yields further improvements, demonstrating that the test-time matching principle generalizes robustly beyond benchmarks with group structures.

**Contributions.** We summarize our main contributions below:

1. **Correcting evaluation metrics.** We show that the widely used evaluation metric GroupScore systematically *underestimates* model capability, and introduce GroupMatch as a faithful measure. We further show that a simple overfitting step translates correctness under GroupMatch into correctness under GroupScore, enabling GPT-4.1 to achieve the first Winoground result surpassing estimated human performance.

2. **Test-time matching for self-improvements.** We propose *Test-Time Matching* (TTM), an iterative, self-improving algorithm that selects matching-induced pseudo-labels for self-training and progressively relaxes the selection threshold to expand coverage. TTM delivers additional, non-trivial

---

[1]We use GPT-4.1-2025-04-14, the most recent GPT model available to us that supports log-probability outputs, enabling more accurate computation of similarity scores (Lin et al., 2024). At the time of writing (September 2025), GPT-5 did not support log-probability outputs.

gains on top of SimpleMatch, enabling SigLIP-B16 to surpass GPT-4.1 on MMVP-VLM and establishing a new state of the art.

3. **Broad applicability of** TTM. We conduct extensive experiments across 16 dataset variants spanning $2 \times 2$, $1 \times k$, and non-grouped settings, demonstrating that TTM consistently improves model performance across diverse scenarios, including those without metric-induced effects or predefined group structures.

**Paper organization.** In Section 2, we review group-structured evaluation for compositional reasoning. In Section 3, we revisit evaluation metrics, introduce a new group matching score (GroupMatch), present our test-time matching (TTM) algorithm, and extend it to global (non-grouped) settings. In Section 4, we report results on benchmarks with $2 \times 2$ groups, $1 \times k$ groups, and non-grouped structures; additional analyses and ablations are provided in Appendix C.2. We conclude in Section 5. Related work, formal proofs, additional experimental details, and extended results are provided in the Appendix.

## 2 PRELIMINARIES

We study compositional reasoning in multimodal models. Benchmarks for this task are typically organized into *groups* of images and captions, often of shape $k \times k$ or $1 \times k$. Within each group, the images and captions differ in subtle yet systematic ways. For example, the widely used Winoground dataset consists of groups with two images and two captions, where both captions contain the same set of words but in different orders, such that each caption correctly describes only one of the two images (Thrush et al., 2022).

To succeed on these benchmarks, a model must correctly align images and captions within each group. Let $s_{ij} := s(I_i, C_j)$ denote the similarity score between image $I_i$ and caption $C_j$. For contrastive vision-language models such as CLIP (Radford et al., 2021) and SigLIP (Zhai et al., 2023), $s_{ij}$ is typically computed as the inner product of image and text embeddings. For multimodal large language models, similarity can instead be estimated using metrics such as VQAScore (Lin et al., 2024). We collect all scores into a similarity matrix $s$, which shares the same shape as the group.

**The** GroupScore **metric for** $k \times k$ **groups.** Consider a group of $k$ images and $k$ captions with ground-truth pairings $\{(I_i, C_i)\}_{i=1}^{k}$ hidden from the learner. The most widely used evaluation metric is the GroupScore (Thrush et al., 2022; Tong et al., 2024; Burapacheep et al., 2024). The GroupScore equals $1$ if the model's similarity scores admit a bijection such that (i) each image is assigned to its correct caption and (ii) each caption is assigned to its correct image; otherwise it equals $0$. Mathematically, we have

$$\mathsf{GroupScore}(s) := \begin{cases} 1 & \forall i : \; s_{ii} > \max_{j \neq i} s_{ij} \quad \text{and} \quad s_{ii} > \max_{j \neq i} s_{ji}, \\ 0 & \text{otherwise.} \end{cases} \tag{1}$$

**Evaluation metrics for** $1 \times k$ **groups.** Without loss of generality, we assume each group consists of 1 image and $k$ captions (Kamath et al., 2023; Hsieh et al., 2023). In this case, the GroupScore reduces to the TextScore, which equals $1$ if the model selects the correct caption and $0$ otherwise.

**Scope and extensions.** In this paper, we primarily focus on $k \times k$ and $1 \times k$ groups as they are the most common configurations in compositional reasoning benchmarks. We defer discussion of general rectangular groups of shape $m \times k$ to Appendix B.2.

## 3 METHODS

In Section 3.1, we show that the standard evaluation metric can systematically *undercount* model capability on group-structured benchmarks, and we introduce a group matching score (GroupMatch) that corrects this artifact. Building on this, we develop an iterative, self-improving *Test-Time Matching* (TTM) algorithm that bootstraps model performance without external supervision (Section 3.2). Finally, we extend TTM beyond group-structured datasets to a global matching formulation applicable to general settings (Section 3.2.1).

### 3.1 REVISITING EVALUATION METRICS: FROM RANDOM GUESSING TO GROUP MATCHING

Most compositional reasoning benchmarks use the GroupScore metric described in Section 2. Despite the broad practical success of frontier AI models, reported results on established benchmarks—particularly those with $k \times k$ groups—are often *at or below random guessing* (Thrush et al., 2022; Diwan et al., 2022; Tong et al., 2024; Burapacheep et al., 2024; Li et al., 2025).[2]

**Revisiting evaluation metrics.** Such counter-intuitive results motivate us to re-examine evaluation metrics for $k \times k$ groups. To calibrate their behavior, we analyze a *random guessing model* under each metric. Consider a group of $k$ images $\{I_i\}_{i=1}^k$ and $k$ captions $\{C_i\}_{i=1}^k$, with ground-truth pairings $\{(I_i, C_i)\}_{i=1}^k$ hidden from the learner (Thrush et al., 2022; Tong et al., 2024; Burapacheep et al., 2024). For each pair $(I_i, C_j)$, the random guessing model assigns a similarity score $\mathsf{sim}(I_i, C_j) \sim \mathrm{unif}([0, 1])$, producing a similarity matrix $s \in \mathbb{R}^{k \times k}$ with entries $s_{ij} := \mathsf{sim}(I_i, C_j)$.

Under the widely used GroupScore metric, achieving a score of $1$ requires the similarity matrix $s$ to satisfy $2k^2 - 2k$ constraints (see Eq. (1)). Equivalently, each diagonal entry $s_{ii}$ must be the largest element in both its row and column—a highly restrictive condition. The probability of achieving a group score of $1$ under random guessing is given below (see Appendix B.1 for proofs).

**Proposition 1.** *For random similarity scores $s \in \mathbb{R}^{k \times k}$, $\mathbb{P}(\mathsf{GroupScore}(s) = 1) = \frac{(k-1)!}{(2k-1)!}$.*

**A new group matching score.** We introduce a new evaluation metric that evaluates the *best overall matching* rather than isolated pairwise comparisons. We consider *bijective matchings* (one-to-one and onto) from images to captions. Let $\pi$ denote such a matching, where $\pi(i)$ is the caption assigned to image $i$. We define the GroupMatch as

$$\mathsf{GroupMatch}(s) := \begin{cases} 1 & \text{if } \sum_{i=1}^k s_{i,\pi^\star(i)} > \sum_{i=1}^k s_{i,\pi(i)}, \quad \forall\, \pi \neq \pi^\star, \\ 0 & \text{otherwise,} \end{cases}$$

where $\pi^\star : i \mapsto i$ denotes the ground-truth matching. Intuitively, the GroupMatch equals $1$ if the *total similarity* of the ground-truth matching exceeds that of all other possible matchings. For $k = 2$, this reduces to the simple condition $s_{11} + s_{22} > s_{12} + s_{21}$. Since there are $k!$ distinct matchings (permutations) and, under random guessing, each is equally likely to maximize the total score, we obtain the following result.

**Proposition 2.** *For random similarity scores $s \in \mathbb{R}^{k \times k}$, $\mathbb{P}(\mathsf{GroupMatch}(s) = 1) = \frac{1}{k!}$.*

**Remark 1.** *The GroupMatch naturally extends to general rectangular groups of shape $m \times k$ (with $m < k$) by considering all injective matchings (one-to-one). In these cases, it also improves over the GroupScore, increasing the expected random guessing score from $1/k^m$ to $(k - m)!/k!$ (see Appendix B.2 for details). In the special case of $1 \times k$ groups, the GroupMatch and the GroupScore coincide.*

SimpleMatch**: translating** GroupMatch **to** GroupScore. Two key observations emerge:

- $\mathbb{P}(\mathsf{GroupMatch}(s) = 1) > \mathbb{P}(\mathsf{GroupScore}(s) = 1)$ for all integers $k > 1$.
- If the correct matching $\pi^\star$ is selected, overfitting to $\pi^\star$ at test time guarantees a GroupScore of $1$. In other words, correctness under GroupMatch can be translated into correctness under GroupScore.

Taken together, these observations imply that GroupScore **systematically underestimates model capability**: it imposes unnecessary constraints and can miscount correct matchings as wrong. In fact, one can easily improve model performance under GroupScore by (i) selecting the most likely matching under GroupMatch and (ii) overfitting to that matching at test time to transfer gains.[3] We refer to this approach as SimpleMatch with GroupMatch. In the commonly studied case with $k = 2$, the expected group score of a random guessing model increases from $1/6$ to $1/2$.

---

[2]These benchmarks are widely adopted; for example, as of October 2025, Winoground (Thrush et al., 2022) has over 500 citations and MMVP-VLM (Tong et al., 2024) has nearly 500 citations.

[3]Since overfitting to matchings induced by GroupMatch achieves the same level of performance under GroupScore, throughout the paper, we report raw model performance under GroupScore and our algorithms' performance under GroupMatch. The latter can always be converted to equivalent GroupScore performance with an additional overfitting step.

---

**Algorithm 1** Test-Time Matching (TTM)

---

**Input:** Pretrained $f_0$; test set of groups $\mathcal{D} = \{G_i\}_{i=1}^n$; number of iterations $T$; thresholds $\{\tau_t\}_{t=1}^T$.

1: **for** iteration $t = 1$ to $T$ **do**
2:     Initialize pseudo-labeled set $\mathcal{S}_t \leftarrow \emptyset$.
3:     **for** each group $G_i \in \mathcal{D}$ **do**
4:         Induce matching $\pi_{f_{t-1}}(G_i) \leftarrow \arg\max_\pi s(\pi; G_i, f_{t-1})$.
5:         Compute margin $\Delta(G_i; f_{t-1})$ as
$$\Delta(G_i; f_{t-1}) \leftarrow s(\pi_{f_{t-1}}(G_i); G_i, f_{t-1}) - \max_{\pi \neq \pi_{f_{t-1}}(G_i)} s(\pi; G_i, f_{t-1}).$$
6:         **if** $\Delta(G_i; f_{t-1}) \geq \tau_t$ **then**
7:             $\mathcal{S}_t \leftarrow \mathcal{S}_t \cup \{(G_i, \pi_{f_{t-1}}(G_i))\}$.
8:     Finetune model on $\mathcal{S}_t$ to obtain $f_t$. `// Self-improving with no external supervision.`
**Output:** Test-time adapted model $f_T$.

---

**Empirical validation.** We evaluate SimpleMatch on SigLIP (Zhai et al., 2023) and GPT-4.1 across three established compositional reasoning benchmarks with $k \times k$ group structures: Winoground (Thrush et al., 2022), MMVP-VLM (Tong et al., 2024), and Colorswap (Burapacheep et al., 2024). Results are presented in Fig. 1. SimpleMatch reveals substantial hidden capability: SigLIP-B16 improves from $10.25 \rightarrow 67$ on Winoground, $22.96 \rightarrow 81.48$ on MMVP-VLM, and $30.33 \rightarrow 88$ on ColorSwap, surpassing all previous results without access to additional data (Wu et al., 2023; Vaishnav & Tammet, 2025; Zhang et al., 2024c; Burapacheep et al., 2024). GPT-4.1 also improves dramatically, from $69.75 \rightarrow 91.38$ on Winoground, $68.15 \rightarrow 88.52$ on MMVP-VLM, and $91.08 \rightarrow 97.42$ on ColorSwap—*yielding the first result to surpass the estimated human performance of 85.5 on Winoground* (Thrush et al., 2022).

## 3.2 Test-Time Matching: iterative bootstrapping of model performance

The GroupMatch metric introduced in Section 3.1 reveals substantial model capability masked by previous metrics. To push performance further, we introduce a test-time matching algorithm that iteratively bootstraps model performance, yielding new state-of-the-art results. Our method applies to groups of general shapes: we consider bijective matchings for square groups and injective matchings for rectangular groups.

**High-level idea.** Our test-time matching algorithm (Algorithm 1) proceeds iteratively for $T$ iterations. At each round $t \in [T]$, the current model $f_{t-1}$ induces candidate matchings for all groups, which serve as pseudo-labels. The algorithm then retains only those matchings it is most confident about, and finetunes on them to obtain the next model $f_t$. By repeating this process, the model progressively self-improves directly at test time, without any external supervision.

The core of Algorithm 1 lies in two design choices: (1) how pseudo-labels are induced within each group, and (2) how the confidence thresholds are scheduled across iterations. We discuss both below.

**Group matching and pseudo-labeling.** For a group $G$ and model $f_{t-1}$, we define the induced matching $\pi_{f_{t-1}}(G) := \arg\max_\pi s(\pi; G, f_{t-1})$, where $s(\pi; G, f_{t-1}) := \sum_u s_{u,\pi(u)}(G; f_{t-1})$ denotes the total similarity of matching $\pi$ on $G$ under $f_{t-1}$. For example, in a $2 \times 2$ group, $\pi_{f_{t-1}}(G) = (1 \mapsto 1, 2 \mapsto 2)$ if $s_{11} + s_{22} > s_{12} + s_{21}$, and $(1 \mapsto 2, 2 \mapsto 1)$ otherwise. For a $1 \times k$ group, the induced matching is $(1 \mapsto \arg\max_{j \in [k]} s_{1j})$. We convert $\pi_{f_{t-1}}(G)$ into a pseudo-label $(G, \pi_{f_{t-1}}(G))$ and add it to the training set $\mathcal{S}_t$ only when its *margin*

$$\Delta(G; f_{t-1}) := s(\pi_{f_{t-1}}(G); G, f_{t-1}) - \max_{\pi \neq \pi_{f_{t-1}}(G)} s(\pi; G, f_{t-1})$$

is greater than or equal to a threshold $\tau_t$. By controlling the threshold, we ensure that the model retains pseudo-labels it is sufficiently confident about.

**A decaying selection threshold schedule.** Any pseudo-label-based method (including TTM) faces two types of pseudo-labeling errors: (i) *false positives*—incorrect pseudo-labels that are mistakenly included in the training set, and (ii) *false negatives*—correct predictions that are excluded due to

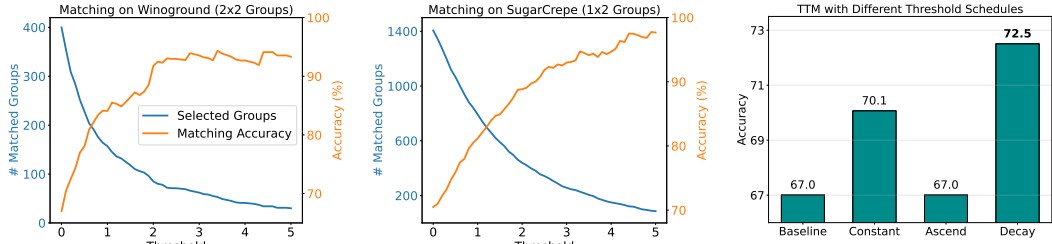

Figure 2: *Left and middle:* Matching results across different thresholds on Winoground and SugarCrepe (the Replace Relation subset) with SigLIP-B16. *Right:* Performance of TTM under different threshold schedules on Winoground with SigLIP-B16. *Baseline* denotes model performance without TTM (under GroupMatch). *Constant* applies TTM with a fixed threshold $\tau_t = 2.0$. *Ascend* applies TTM with a linearly increasing schedule from $\tau_1 = 0$ to $\tau_T = 2.0$, but yields no gains as the model quickly overfits to all pseudo-labels in the first iteration. *Decay* applies TTM with a linearly decreasing schedule from $\tau_1 = 2.0$ to $\tau_T = 0$, yielding the best performance.

overly strict selection criteria. In matching-based pseudo-labeling, a lower threshold $\tau$ increases the number of selected pseudo-labels but typically reduces precision (more false positives), whereas a higher threshold yields cleaner but fewer pseudo-labels (more false negatives). This trade-off is illustrated in the left and middle plots of Fig. 2, which report the number of matched groups (blue) and the accuracy among matched groups (orange) as a function of $\tau$.

TTM employs a *decaying* threshold schedule ($\tau_{t+1} < \tau_t$) to balance these errors over iterations: it begins with a high threshold to ensure high-precision pseudo-labels (few false positives), and then gradually lowers the threshold to increase coverage and reduce false negatives as the model improves. In contrast, an *ascending* schedule can admit substantial false positives in early rounds, which derails learning and limits subsequent improvement; a *constant* schedule avoids early false positives by keeping the threshold high, but induces many false negatives in later iterations, leading to limited gains and early plateauing. The right plot of Fig. 2 supports this intuition: the decaying schedule consistently outperforms the alternatives.

In practice, we find it effective to set the initial threshold $\tau_1$ such that roughly 15%–30% of the groups are matched, and the final threshold $\tau_T$ such that more than 90% of the test set is covered. Both cosine and linear decay schedules perform well. Further analyses and ablations are provided in Appendix C.2.

**Runtime analysis.** The runtime of TTM scales as $O(T \cdot C_{\text{ft}})$, where $T$ is the number of iterations and $C_{\text{ft}}$ denotes the per-iteration model finetuning cost. Empirically, TTM demonstrates strong improvements even with a small number of iterations (e.g., $T = 3$ or $10$). Thus, the runtime of TTM is comparable to standard test-time training methods in the literature (Sun et al., 2020; Akyürek et al., 2025). See Appendix C.4 for a detailed analysis and discussion.

### 3.2.1 TEST-TIME MATCHING WITHOUT GROUP STRUCTURES

While Algorithm 1 is designed for datasets organized into local groups, the same principle extends naturally to settings without any predefined group structure. In this case, we treat the entire dataset as a single global matching problem between all images and all captions.

Let $\mathcal{S}_I$ denote the set of images and $\mathcal{S}_C$ the set of captions. We assume $|\mathcal{S}_I| \le |\mathcal{S}_C|$ and each image has a unique corresponding caption (one-to-one assignment). Let $s \in \mathbb{R}^{|\mathcal{S}_I| \times |\mathcal{S}_C|}$ be the similarity matrix produced by a model $f$. We consider all injective matchings $\pi : \mathcal{S}_I \to \mathcal{S}_C$ from images to captions. The model-induced global matching is then defined as

$$\pi_f := \arg\max_{\pi : \mathcal{S}_I \to \mathcal{S}_C} \sum_{i \in \mathcal{S}_I} s_{i, \pi(i)}, \tag{2}$$

which maximizes the total similarity over image-caption pairs. Eq. (2) corresponds to the classical *assignment problem*, which can be efficiently solved by strongly-polynomial time algorithms such as the Hungarian algorithm (Kuhn, 1955).

Analogous to Algorithm 1, we adopt an iterative schedule with pseudo-labeling. At iteration $t$, let $\pi_{f_{t-1}}$ be the global matching induced by model $f_{t-1}$. Because the entire dataset is treated as a single group, group-level margin thresholding loses granularity: the model would either accept all matches or none. To address this, we apply thresholding at the level of individual pairs. Specifically, the pseudo-label set at iteration $t$ is

$$\mathcal{S}_t := \{(i, \pi_{f_{t-1}}(i)) : s_{i,\pi_{f_{t-1}}(i)} \geq \tau_t\},$$

where $\tau_t$ is the threshold at iteration $t$. The threshold can be set either as an absolute value or relative to the distribution of similarity scores (i.e., the $p$-th percentile). Following the same principle as in Algorithm 1, we begin with a relatively high threshold to ensure high-precision pseudo-labels and gradually decay it over iterations to expand coverage and bootstrap performance over the test set.

## 4 EXPERIMENTS

We describe the experimental setups in Section 4.1, present the main results in Sections 4.2 to 4.5, and provide analyses and ablations in Appendix C.2. Additional experimental details and results are deferred to Appendix C.

### 4.1 EXPERIMENTAL SETUPS

**Datasets.** We evaluate on five challenging compositional reasoning benchmarks: Winoground (Thrush et al., 2022), MMVP-VLM (Tong et al., 2024), Colorswap (Burapacheep et al., 2024), SugarCrepe (Hsieh et al., 2023), and WhatsUp (Kamath et al., 2023). Winoground, MMVP-VLM, and Colorswap consist of $2 \times 2$ groups; we also construct their non-grouped variants by discarding group structures (Section 3.2.1). SugarCrepe consists of $1 \times 2$ groups and WhatsUp consists of $1 \times 4$ groups; we evaluate on 4 different subsets of SugarCrepe and all 2 subsets of WhatsUp. Following Li et al. (2025), we further convert WhatsUp into 4 different variants with $2 \times 2$ groups. In total, our evaluation spans 16 dataset variations covering diverse structures and evaluation settings.

**Models.** We test both contrastive vision-language models and multimodal large language models. For contrastive models, we use SigLIP (Zhai et al., 2023) and CLIP (Radford et al., 2021) at multiple scales, including SigLIP-B16, SigLIP-L16, CLIP-B16, and CLIP-B32. For multimodal large language models, we use GPT-4.1, where image-text similarity is computed based on VQAScore (Lin et al., 2024).

**Evaluation metrics.** For GPT-4.1, we report raw GroupScore and GroupMatch-induced performance via SimpleMatch (Section 3.1). For CLIPs and SigLIPs, we additionally include results with TTM (Algorithm 1). Specifically: on $2 \times 2$ datasets we report (i) raw GroupScore, (ii) GroupMatch-induced performance, and (iii) TTM-boosted performance; on $1 \times k$ datasets we report (i) raw GroupScore and (ii) TTM-boosted performance, since GroupScore and GroupMatch coincide in this case; and on datasets without group structures we report (i) raw GroupScore (with known groups), (ii) global assignment accuracy under Eq. (2), and (iii) TTM-boosted performance via the global variant introduced in Section 3.2.1. In all cases, we highlight performance gains from TTM—over GroupMatch for $2 \times 2$ datasets, over GroupScore for $1 \times k$ datasets, and over global assignment accuracy under Eq. (2) for datasets without group structures. All results are averaged over four random runs, with standard deviations reported.

### 4.2 TTM ACHIEVES NEW SOTAS

We evaluate on three established compositional reasoning benchmarks—Winoground, MMVP-VLM, and ColorSwap—all consisting of $2 \times 2$ groups and considered challenging for frontier AI models. Previous state-of-the-art results include 58.75 on Winoground (GPT-4V with prompt tuning (Wu et al., 2023; Vaishnav & Tammet, 2025)), 70.7 on MMVP (via a GPT-4o multi-agent system with tool use (Zhang et al., 2024c)),[4] and 87.33 on ColorSwap without training-set access (95.33 with finetuning on the training set (Burapacheep et al., 2024)).

---

[4]This result is on MMVP, a variant of MMVP-VLM formulated as binary-choice question answering. In this paper, we focus on MMVP-VLM, which is better suited for contrastive models. Prior work has shown that model performance on the two variants is positively correlated (Tong et al., 2024; Li et al., 2025).

Table 1: Performance on Winoground, MMVP-VLM, and ColorSwap. Raw model performance is reported under GroupScore. SimpleMatch corresponds to the performance under GroupMatch (Section 3.1), and TTM corresponds to the performance of Algorithm 1. We report absolute gains ($\Delta$), relative gains, and relative error reductions of TTM over SimpleMatch. Cells highlighted in ▨ indicate results obtained with TTM, while cells in ▨ denote the **SOTA** performance for each dataset.

| Dataset / Model | Raw | SimpleMatch | TTM | $\Delta$ | | Error Red. |
|---|---|---|---|---|---|---|
| **Winoground** | | | | | | |
| GPT-4.1 | $69.75_{\pm 0.56}$ | $91.38_{\pm 0.80}$ | – | – | | – |
| CLIP-B16 | 7.25 | 60.00 | $\mathbf{65.44}_{\pm 1.10}$ | $\mathbf{+5.4}$ | $(9.1\% \uparrow)$ | $\mathbf{13.6\%} \downarrow$ |
| SigLIP-B16 | 10.25 | 67.00 | $\mathbf{72.50}_{\pm 0.64}$ | $\mathbf{+5.5}$ | $(8.2\% \uparrow)$ | $\mathbf{16.7\%} \downarrow$ |
| SigLIP-L16 | 13.00 | 69.50 | $\mathbf{72.75}_{\pm 0.64}$ | $\mathbf{+3.3}$ | $(4.7\% \uparrow)$ | $\mathbf{10.7\%} \downarrow$ |
| **MMVP-VLM** | | | | | | |
| GPT-4.1 | $68.15_{\pm 0.00}$ | $88.52_{\pm 0.83}$ | – | – | | – |
| CLIP-B16 | 5.19 | 72.59 | $\mathbf{80.19}_{\pm 0.81}$ | $\mathbf{+7.6}$ | $(10.5\% \uparrow)$ | $\mathbf{27.7\%} \downarrow$ |
| SigLIP-B16 | 22.96 | 81.48 | $\mathbf{89.44}_{\pm 0.96}$ | $\mathbf{+8.0}$ | $(9.8\% \uparrow)$ | $\mathbf{43.0\%} \downarrow$ |
| **ColorSwap** | | | | | | |
| GPT-4.1 | $91.08_{\pm 0.28}$ | $97.42_{\pm 0.14}$ | – | – | | – |
| CLIP-B16 | 12.00 | 77.67 | $\mathbf{85.75}_{\pm 0.64}$ | $\mathbf{+8.1}$ | $(10.4\% \uparrow)$ | $\mathbf{36.2\%} \downarrow$ |
| SigLIP-B16 | 30.33 | 88.00 | $\mathbf{94.25}_{\pm 0.43}$ | $\mathbf{+6.3}$ | $(7.1\% \uparrow)$ | $\mathbf{52.1\%} \downarrow$ |
| SigLIP-L16 | 37.00 | 91.33 | $\mathbf{96.08}_{\pm 0.43}$ | $\mathbf{+4.8}$ | $(5.2\% \uparrow)$ | $\mathbf{54.8\%} \downarrow$ |

**Simple matching reveals hidden capabilities.** Applying SimpleMatch (Section 3.1) to CLIP, SigLIP, and GPT-4.1 already yields striking improvements (Table 1). SimpleMatch enables SigLIP-B16 to surpass all prior state-of-the-art results without access to additional data, and enables GPT-4.1 to set new records across all three benchmarks. Notably, GPT-4.1 improves from 69.75 to 91.38 on Winoground, *yielding the first result to surpass the estimated human performance of 85.5* (Thrush et al., 2022). These findings confirm that the GroupMatch metric can reveal substantial hidden compositional reasoning capabilities.

**Test-time matching further boosts performance.** We next apply TTM (Algorithm 1) to CLIP and SigLIP, enabling additional performance gains without external supervision. As shown in Table 1, TTM *consistently improves over* SimpleMatch *across datasets and model scales, with relative gains up to 10.5% and relative error reduction up to 54.8%.*[5] Crucially, TTM elevates SigLIP-L16 to the level of GPT-4.1 on ColorSwap and **enables SigLIP-B16 to surpass GPT-4.1 on MMVP-VLM, establishing a new state of the art**. These results demonstrate that TTM is a powerful and practical approach for enhancing model performance through self-improvement at test time.

### 4.3 TTM ALSO IMPROVES GENERATIVE MULTIMODAL MODELS

Beyond contrastive vision-language models, we also apply TTM (Algorithm 1) to a small yet capable generative multimodal model, SmolVLM-256M-Instruct (Marafioti et al., 2025). We compute the similarity score using VQAScore (Lin et al., 2024) with the prompt: <image> Does this image show "<text>"? Please answer "Yes" or "No". During training, the pseudo-labeled dataset includes both "Yes" and "No" responses. The "No" examples are constructed from mismatched image-caption pairs within the same group, providing hard negatives without requiring additional data sources. Results are provided in Table 2. Although these benchmarks are primarily designed for contrastive vision-language models (e.g., their short captions are not always natural for generative MLLMs), TTM still yields clear gains over SimpleMatch, with the significant improvements on MMVP-VLM and ColorSwap.

---

[5]While the absolute boosts may appear modest compared to GroupMatch-induced gains, they are *highly significant*: for comparison, scaffolding GPT-4V yields only a 1.25-point gain on the Winoground dataset, improving performance from 50.75 (Zhang et al., 2024a) to 52 (Vaishnav & Tammet, 2025).

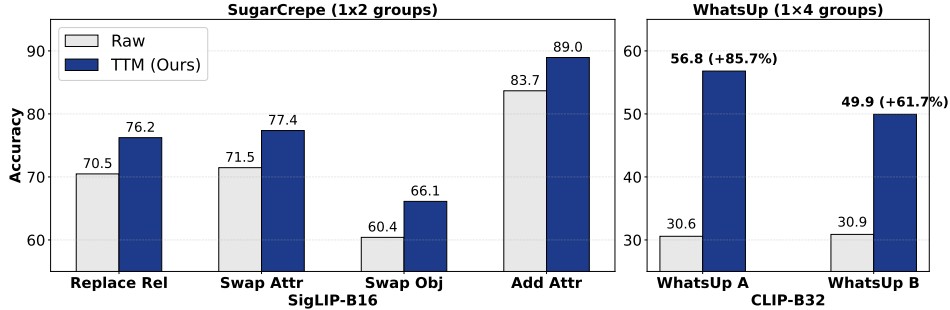

Figure 3: TTM results on benchmarks without metric-induced boosts: for $1 \times k$ groups, GroupMatch (and thus SimpleMatch) coincide with GroupScore. *Left:* results on four SugarCrepe subsets consisting of $1 \times 2$ groups. *Middle:* results on both WhatsUp subsets consisting of $1 \times 4$ groups.

Table 2: Performance of SmolVLM-256M-Instruct on Winoground, MMVP-VLM, and ColorSwap. Raw model performance is reported under GroupScore, SimpleMatch corresponds to the performance under GroupMatch (Section 3.1), and TTM corresponds to the performance of Algorithm 1. We report absolute gains ($\Delta$), relative gains, and relative error reduction of TTM over SimpleMatch.

| Datasets | SmolVLM | SimpleMatch | + TTM | $\Delta$ | | Error Red. |
|---|---|---|---|---|---|---|
| Winoground | 7.25 | 61.75 | $\mathbf{63.38}_{\pm 0.67}$ | $\mathbf{+1.6}$ | $\mathbf{(2.6\% \uparrow)}$ | $\mathbf{4.3\% \downarrow}$ |
| MMVP-VLM | 20.00 | 76.30 | $\mathbf{81.67}_{\pm 1.52}$ | $\mathbf{+5.4}$ | $\mathbf{(7.0\% \uparrow)}$ | $\mathbf{22.7\% \downarrow}$ |
| ColorSwap | 30.00 | 80.00 | $\mathbf{85.17}_{\pm 1.09}$ | $\mathbf{+5.2}$ | $\mathbf{(6.5\% \uparrow)}$ | $\mathbf{25.9\% \downarrow}$ |

## 4.4 TTM REMAINS EFFECTIVE WITHOUT METRIC-INDUCED BOOSTS

To evaluate the effectiveness of Algorithm 1 beyond cases where alternative metrics can inflate performance, we consider benchmarks with $1 \times k$ group structure, where GroupScore and GroupMatch coincide and thus provide no metric-induced boost.

We experiment on 4 SugarCrepe subsets ($1 \times 2$ groups) and all 2 WhatsUp subsets ($1 \times 4$ groups), reporting results in Fig. 3. Even without metric-induced gains, Algorithm 1 consistently delivers substantial test-time improvements. The gains are especially striking on the WhatsUp datasets, where **performance improves by up to 85.7%**, turning these previously challenging tasks into tractable ones.

Following Li et al. (2025), we further convert the WhatsUp datasets into 4 directional variants with $2 \times 2$ group structures. As shown in Table 10 (in Appendix C.3), Algorithm 1 again yields significant improvements—**up to 135.1% relative gains and 95.5% relative error reduction**—on top of SimpleMatch. Together, these results demonstrate that TTM is broadly effective across both $k \times k$ and $1 \times k$ groups, even when metric-induced effects are absent, as in the case of $1 \times k$ groups.

## 4.5 TTM EXTENDS BEYOND GROUP-STRUCTURED DATASETS

To further assess the generality of Algorithm 1, we evaluate its global variant introduced in Section 3.2.1 on datasets *without any predefined group structures*. Specifically, we flatten Winoground, MMVP-VLM, and ColorSwap by removing local $k \times k$ groups, resulting in a general dataset with an image set $\mathcal{S}_I$ and a caption set $\mathcal{S}_C$.

We report three metrics: (i) raw GroupScore (with the extra knowledge of the group structure), (ii) global assignment accuracy obtained via SimpleMatch under Eq. (2), and (iii) TTM-boosted performance achieved using the global variant introduced in Section 3.2.1. Results show that even global assignment without group structures substantially outperforms the vanilla GroupScore, demonstrating the effectiveness of using *matching-based supervision* to generate high-quality pseudo-labels. More importantly, applying the iterative global TTM algorithm yields further gains over global assignment alone, with especially large relative error reductions on ColorSwap, i.e., **33.3%**

Table 3: Performance on non-grouped variants of Winoground, MMVP-VLM, and ColorSwap. Raw model performance is reported under GroupScore, SimpleMatch corresponds to the performance of global assignment defined in Eq. (2), and TTM corresponds to the performance of the global variant of Algorithm 1. We report absolute gains ($\Delta$), relative gains, and relative error reduction of TTM over SimpleMatch.

| Datasets | SigLIP-B16 | SimpleMatch | + TTM | $\Delta$ | Error Red. |
|---|---|---|---|---|---|
| Winoground | 10.25 | 44.38 | $\textbf{46.78}_{\pm 1.05}$ | +2.4 (5.4% ↑) | 4.3% ↓ |
| MMVP-VLM | 22.96 | 39.63 | $\textbf{44.54}_{\pm 2.02}$ | +4.9 (12.4% ↑) | 8.1% ↓ |
| ColorSwap | 30.33 | 88.00 | $\textbf{92.00}_{\pm 1.24}$ | +4.0 (4.5% ↑) | 33.3% ↓ |

**relative error reduction on ColorSwap** (see Table 3). This demonstrates that the test-time matching principle generalizes effectively beyond group-structured datasets.

## 5  DISCUSSION

This work revisits the long-standing puzzle of compositional reasoning, where modern multimodal models often appear to perform no better than random guessing (Thrush et al., 2022; Diwan et al., 2022; Tong et al., 2024; Burapacheep et al., 2024; Li et al., 2025). We show that this apparent limitation partly arises from the evaluation metrics themselves, which systematically underestimate model capability. We introduce GroupMatch as a metric correction that yields a more faithful evaluation and can be translated back to the standard metric via a simple overfitting step; under this correction, GPT-4.1 surpasses estimated human performance on Winoground. Building on this insight, we propose *Test-Time Matching* (TTM), an iterative, self-improving algorithm that further bootstraps model performance without external supervision. TTM enables SigLIP-B16 to outperform GPT-4.1 on MMVP-VLM, establishing a new state of the art. Experiments across 16 dataset variants demonstrate that TTM consistently improves performance across diverse settings, including those without metric-induced effects or predefined group structures.

Moving forward, we highlight two promising directions:

- **Recalibrating model evaluation.** The same model on the same dataset can yield vastly different results under different metrics. This underscores the need for more robust, transparent, and reliable evaluation protocols for compositional reasoning and beyond (Schaeffer et al., 2023).

- **Extending** TTM **beyond compositional reasoning.** While developed in the context of compositional reasoning, the core principle of TTM—iterative, matching-based self-training at test time—is general. A natural next step is to explore this idea in broader multimodal or language settings.

AUTHOR CONTRIBUTIONS

YZ conceived the project, developed the algorithms, performed the majority of the implementation and experiments, and wrote the manuscript. JZ and FT assisted with the implementation; JZ additionally conducted experiments on the WhatsUp datasets.

ACKNOWLEDGMENTS

YZ acknowledges support from NSF IIS 2425006.

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

## A  RELATED WORK

**Compositional reasoning and evaluation metrics.**  Contrastive vision-language models (VLMs) such as CLIP (Radford et al., 2021) and SigLIP (Zhai et al., 2023), and multimodal large language models (MLLMs) such as the GPT (Achiam et al., 2023; Hurst et al., 2024) and Gemini (Team et al., 2023; Comanici et al., 2025) series, have achieved remarkable progress across a wide range of multimodal tasks. Yet both VLMs and MLLMs struggle on benchmarks specifically designed to test *compositional reasoning*—the ability to systematically combine objects, attributes, and relations to interpret or reason about novel configurations (Lake et al., 2017; Bahdanau et al., 2019; Thrush et al., 2022; Hsieh et al., 2023; Kamath et al., 2023; Tong et al., 2024; Burapacheep et al., 2024). These benchmarks are typically organized into small groups of images and captions that differ in subtle but systematic ways (e.g., captions with identical words but different orderings). The prevailing evaluation metric, the GroupScore, requires models to correctly assign each image to its corresponding caption and each caption to its corresponding image via isolated pairwise comparisons. While rigorous, this metric is also unforgiving: raw model performance often falls at or below random guessing (Thrush et al., 2022; Diwan et al., 2022; Tong et al., 2024; Burapacheep et al., 2024; Li et al., 2025).

Despite recent attempts to improve compositional reasoning in frontier multimodal models (Wu et al., 2023; Zhang et al., 2024c; Vaishnav & Tammet, 2025), progress remains modest. For instance, the previous state of the art on Winoground—achieved by scaffolding and prompt tuning GPT-4V (Wu et al., 2023; Vaishnav & Tammet, 2025)—was only 58.75, still well below the estimated human performance of 85.5 (Thrush et al., 2022).

Our work takes a complementary perspective to prior efforts by revisiting the evaluation metrics used in compositional reasoning. We introduce a *group matching score* (GroupMatch) that evaluates the best overall matching rather than isolated pairwise comparisons, revealing substantial hidden capability in both VLMs and MLLMs. Crucially, by simply overfitting to the induced matchings at test time, this hidden capability transfers into higher scores under the original GroupScore, closing much of the reported gap. With this adjustment, GPT-4.1 improves from 69.75 to 91.38 on Winoground— *yielding the first result to surpass the estimated human performance of 85.5*. This finding echoes broader observations that measured capability can be highly sensitive to the choice of evaluation metric (Schaeffer et al., 2023), underscoring the need for continued research on evaluation protocols for frontier models.

**Test-time training, pseudo-labeling, and adaptive schedules.**  Test-time training adapts models during inference to improve performance, with roots in early work on local learning and instance-specific adaptation (Cleveland, 1979; Cleveland & Devlin, 1988; Bottou & Vapnik, 1992; Atkeson et al., 1997). The idea has regained attention in the era of large pretrained models, where test-time self-supervision can enhance performance without additional labeled data (Sun et al., 2020; Wang et al., 2020; Gandelsman et al., 2022; Chen et al., 2022). Recent studies show that finetuning on retrieved data based on test prompts can significantly improve large language models (Hardt & Sun, 2024; Hübotter et al., 2025), and test-time training has become a key component in tackling reasoning-heavy benchmarks such as ARC (Chollet, 2019; Chollet et al., 2024; Akyürek et al., 2025).

Our test-time matching algorithm (TTM) shares this motivation but differs in key aspects. Most prior methods adapt to each test instance independently, producing per-instance finetuned models and often relying on instance-specific in-context examples (Akyürek et al., 2025). In contrast, TTM leverages GroupMatch-induced pseudo-labels across the *entire test set*, iteratively updating a single model through an adaptive thresholding schedule. This connects naturally to the literature on self-training (Kumar et al., 2020) and semi-supervised learning (Zhu, 2005; Chapelle et al., 2009; Sohn et al., 2020; Zhang et al., 2021, 2024b), where pseudo-labels drive improvements. A central contribution of our approach is to exploit matching and group structure—both locally (Section 3.2) and globally (Section 3.2.1)—to generate high-quality pseudo-labels.

Finally, our adaptive thresholding schedule resonates with classical ideas in active learning (Castro & Nowak, 2007; Balcan et al., 2007; Dasgupta et al., 2009; Hanneke, 2014; Krishnamurthy et al., 2019; Puchkin & Zhivotovskiy, 2021; Zhu & Nowak, 2022a,b), though with reversed logic: whereas active learning typically queries the most uncertain examples for human annotation, our approach begins with the most confident pseudo-labels and gradually relaxes thresholds to expand coverage. This

confidence-first perspective is central to the effectiveness of TTM, enabling consistent performance gains without any external supervision.

# B    PROOFS AND SUPPORTING RESULTS FROM SECTION 3

## B.1    PROOFS OF PROPOSITION 1 AND PROPOSITION 2

**Proposition 1.** *For random similarity scores $s \in \mathbb{R}^{k \times k}$, $\mathbb{P}(\mathsf{GroupScore}(s) = 1) = \frac{(k-1)!}{(2k-1)!}$.*

*Proof.* Because the entries of $s$ are i.i.d. sampled from a continuous distribution (here $\mathrm{unif}([0,1])$), ties occur with probability 0, so we may use strict inequalities throughout.

Denote $d_i := s_{ii}$ and, for $i \neq j$, set $m_{ij} := \min\{d_i, d_j\}$. By the definition of the GroupScore, the event $\{\mathsf{GroupScore}(s) = 1\}$ is equivalent to requiring $s_{ij} < m_{ij}$ and $s_{ji} < m_{ij}$ for every $i \neq j$. Conditioning on the diagonal $d = (d_1, \ldots, d_k)$ and using independence of the off-diagonal entries,

$$\mathbb{P}\big(\mathsf{GroupScore}(s) = 1 \mid d\big) = \prod_{i<j} \mathbb{P}(s_{ij} < m_{ij})\, \mathbb{P}(s_{ji} < m_{ij}) = \prod_{i<j} m_{ij}^2.$$

Let $0 < x_1 < \cdots < x_k < 1$ be the order statistics of $(d_1, \ldots, d_k)$. We then have $m_{ij} = x_{\min\{r(i), r(j)\}}$, where $r(\cdot)$ is the rank, hence

$$\prod_{i<j} m_{ij}^2 = \prod_{a=1}^{k} x_a^{2(k-a)}.$$

Since $(x_1, \ldots, x_k)$ are the order statistics of i.i.d. $\mathrm{unif}([0,1])$ samples, their joint density is $k!$ on the ordered region $\{0 < x_1 < \cdots < x_k < 1\}$ (and 0 elsewhere). Therefore,

$$\mathbb{P}\big(\mathsf{GroupScore}(s) = 1\big) = k! \int_{0 < x_1 < \cdots < x_k < 1} \prod_{a=1}^{k} x_a^{2(k-a)}\, dx_1 \cdots dx_k.$$

For $1 \leq \ell \leq k$ and $y \in (0, 1]$, define

$$I_\ell(y) := \int_{0 < x_1 < \cdots < x_\ell < y} \prod_{a=1}^{\ell} x_a^{2(k-a)}\, dx_1 \cdots dx_\ell.$$

We claim that, for $\ell = 1, \ldots, k$,

$$I_\ell(y) = \frac{y^{\ell(2k-\ell)}}{\prod_{r=1}^{\ell} r(2k-r)}.$$

This is proved by induction on $\ell$. For $\ell = 1$,

$$I_1(y) = \int_0^y x^{2(k-1)}\, dx = \frac{y^{2k-1}}{2k-1}.$$

Assume it holds for $\ell - 1$. Then

$$\begin{aligned} I_\ell(y) &= \int_0^y x_\ell^{2(k-\ell)}\, I_{\ell-1}(x_\ell)\, dx_\ell \\ &= \frac{1}{\prod_{r=1}^{\ell-1} r(2k-r)} \int_0^y x_\ell^{2(k-\ell)+(\ell-1)(2k-(\ell-1))}\, dx_\ell \\ &= \frac{1}{\prod_{r=1}^{\ell-1} r(2k-r)} \cdot \frac{y^{\ell(2k-\ell)}}{\ell(2k-\ell)}, \end{aligned}$$

since $2(k-\ell) + (\ell-1)(2k-(\ell-1)) = \ell(2k-\ell) - 1$. Thus the claim holds. Taking $\ell = k$ and $y = 1$ gives

$$\int_{0 < x_1 < \cdots < x_k < 1} \prod_{a=1}^{k} x_a^{2(k-a)}\, dx_1 \cdots dx_k = I_k(1) = \frac{1}{\prod_{r=1}^{k} r(2k-r)}.$$

Therefore,

$$\mathbb{P}\big(\mathsf{GroupScore}(s) = 1\big) = k! \prod_{r=1}^{k} \frac{1}{r(2k-r)} = \frac{(k-1)!}{(2k-1)!}.$$

$\square$

**Proposition 2.** *For random similarity scores $s \in \mathbb{R}^{k \times k}$, $\mathbb{P}(\mathsf{GroupMatch}(s) = 1) = \frac{1}{k!}$.*

*Proof.* There are $k!$ distinct injective matchings. Since the random variables $\{s_{ij}\}$ are continuous, ties occur with probability 0. By symmetry, each injective matching is equally likely to achieve the maximum total similarity. Hence, the probability that the ground-truth matching $\pi^\star$ attains the maximum is $\frac{1}{k!}$. $\square$

## B.2 SUPPORTING RESULTS FOR GENERAL RECTANGULAR GROUPS

Without loss of generality, we consider a group of $m$ images $\{I_i\}_{i=1}^{m}$ and $k$ captions $\{C_i\}_{i=1}^{k}$ with $m < k$. We assume the ground-truth pairings is $\{(I_i, C_i)\}_{i=1}^{m}$ (hidden from the learner). As in the main text, we study a random guessing model that assigns i.i.d. similarity scores $s_{ij} := \mathsf{sim}(I_i, C_j) \sim \mathrm{unif}([0,1])$ for each pair $(I_i, C_j)$, and collect them into a similarity matrix $s \in \mathbb{R}^{m \times k}$.

**GroupScore for $m \times k$ groups.** Analogous to the $k \times k$ and $1 \times k$ cases, the GroupScore for $m \times k$ groups can be defined as

$$\mathsf{GroupScore}(s) := \begin{cases} 1 & \forall\, i \in [m] : \ s_{ii} > \max_{j \neq i} s_{ij}, \\ 0 & \text{otherwise.} \end{cases}$$

Under the random guessing model, the probability of achieving a GroupScore of 1 for rectangular group is given below.

**Proposition 3.** *For random similarity score $s \in \mathbb{R}^{m \times k}$, $\mathbb{P}(\mathsf{GroupScore}(s) = 1) = \frac{1}{k^m}$.*

*Proof.* Since the random variables $\{s_{ij}\}$ are continuous, ties occur with probability 0. For each row $i$, by symmetry, the probability that $s_{ii}$ is the largest among the $k$ i.i.d. entries $\{s_{ij}\}_{j=1}^{k}$ is $1/k$. Since rows are independent, we have

$$\mathbb{P}(\forall\, i \in [m] : s_{ii} > \max_{j \neq i} s_{ij}) = \prod_{i=1}^{m} \frac{1}{k} = \frac{1}{k^m}.$$

$\square$

**GroupMatch for $m \times k$ groups.** We extend GroupMatch to the general rectangular case by considering *injective* matchings $\pi : [m] \to [k]$ (i.e., $\pi(i) \neq \pi(j)$ for $i \neq j$). With the ground-truth injective matching $\pi^\star : i \mapsto i$, we define GroupMatch as

$$\mathsf{GroupMatch}(s) := \begin{cases} 1 & \text{if } \sum_{i=1}^{m} s_{i,\pi^\star(i)} > \sum_{i=1}^{m} s_{i,\pi(i)}, \quad \forall\, \pi \neq \pi^\star, \\ 0 & \text{otherwise.} \end{cases}$$

Under the random guessing model, the probability of achieving a GroupMatch of 1 for rectangular group is given below.

**Proposition 4.** *For random similarity scores $s \in \mathbb{R}^{m \times k}$, $\mathbb{P}(\mathsf{GroupMatch}(s) = 1) = \frac{(k-m)!}{k!}$.*

*Proof.* There are $\frac{k!}{(k-m)!}$ distinct injective matchings. Since the random variables $\{s_{ij}\}$ are continuous, ties occur with probability 0. By symmetry, each injective matching is equally likely to achieve the maximum total similarity. Hence, the probability that the ground-truth matching $\pi^\star$ attains the maximum is $\left(\frac{k!}{(k-m)!}\right)^{-1} = \frac{(k-m)!}{k!}$. $\square$

GroupMatch **helps for rectangular groups.** For random similarity scores $s \in \mathbb{R}^{m \times k}$,

$$\mathbb{P}\big(\text{GroupMatch}(s) = 1\big) = \frac{(k-m)!}{k!} = \frac{1}{k(k-1)\cdots(k-m+1)} \geq \frac{1}{k^m} = \mathbb{P}\big(\text{GroupScore}(s) = 1\big),$$

with strict inequality for any $m \geq 2$ and equality for $m = 1$ (GroupMatch and GroupScore coincide when $m = 1$). Moreover, if the ground-truth injective matching $\pi^\star$ is identified, overfitting to the matching $\pi^\star$ at test time guarantees a GroupScore of 1. Thus, as in the square case, one can improve model performance under GroupScore via SimpleMatch: (i) selecting the most likely matching under GroupMatch and (ii) overfitting to the matching at test time to transfer gains.

## C  ADDITIONAL EXPERIMENTAL DETAILS AND RESULTS

### C.1  IMPLEMENTATION DETAILS AND HYPERPARAMETERS

#### C.1.1  GENERAL TRAINING HYPERPARAMETERS

We summarize the general training setup and hyperparameter choices used in our experiments below. Across all experiments, we use AdamW (Loshchilov & Hutter, 2017) with $(\beta_1, \beta_2) = (0.9, 0.999)$ and weight decay 0.05. The learning rate follows a cosine decay schedule and is restarted at each iteration with a multiplicative factor of 0.95. Optimizer states are reset at each restart, with the exception of SigLIP-B16 on Winoground.

- For contrastive vision-language models, we set the number of iterations to $T = 10$ by default; additional results with fewer iterations are reported in Appendix C.4. At each iteration, we train for 20 epochs by default, except on Winoground, where we train for 30 epochs. We use a batch size of 50 for $2 \times 2$ datasets and 100 for $1 \times k$ datasets; the batch size is defined at the group level (e.g., 50 groups of size $2 \times 2$ per batch).[6]

- For generative multimodal models, we set the number of iterations to $T = 3$ and train for 10 epochs per iteration. We use a batch size of 16 for MMVP-VLM and ColorSwap datasets and a batch size of 32 for Winoground. The batch size here is also defined at the group level. Since the pseudo-labeled dataset includes both "Yes" and "No" responses (Section 4.3), each $2 \times 2$ group provides 4 data points for training.

By default, we do not apply data augmentation during training, as many datasets are designed to be sensitive to location or color. However, we find it beneficial to apply a simple resizing (factor 1.1) followed by random cropping for the following dataset-model pairs: Winoground with SigLIP-L16, MMVP-VLM with SigLIP-B16, ColorSwap with SigLIP-B16, MMVP-VLM with SigLIP-B16 under global matching, and CLIP-B32 with WhatsUp A-Left-Right.

#### C.1.2  TTM-SPECIFIC HYPERPARAMETERS

In Tables 4 to 7, we report, for each dataset-model pair, the initial threshold $\tau_1$, the final threshold $\tau_T$, the threshold decay schedule (linear or cosine), and the learning rate (lr). For group matching (Tables 4 to 6), we use absolute thresholds. For global matching (Table 7), we adopt the percentile-based thresholding mentioned in Section 3.2.1: at iteration $t$, the top $1 - \tau_t$ fraction of pseudo-labels (ranked by similarity) is selected.

In our experiments, the final threshold $\tau_T$ is set to either $0$ (full coverage) or $0.1$ (typically covering more than 90% of the data). The initial threshold $\tau_1$ is more dataset- and model-dependent. For group matching, we find it effective to set $\tau_1$ such that roughly 15%–30% of the groups are matched initially, though in some cases we use thresholds outside this range when they yield better performance (e.g., higher selection fractions for ColorSwap with SigLIP models and lower fractions for WhatsUp $2 \times 2$ variants with CLIP-B32). For global matching, performance tends to improve with a larger initial selection fraction—typically around 50%.

---

[6]We slightly increase the batch size when the total number of groups is just above a multiple of the default size. For instance, if the dataset contains 102 groups, we set the batch size to 51.

Table 4: Hyperparameters used for experiments in Section 4.2.

| Dataset | Model | $\tau_1$ | $\tau_T$ | Schedule | lr |
|---|---|---|---|---|---|
| Winoground | CLIP-B16 | 0.9 | 0 | linear | $2.0 \times 10^{-5}$ |
| | SigLIP-B16 | 2.0 | 0 | linear | $1.0 \times 10^{-5}$ |
| | SigLIP-L16 | 2.0 | 0.1 | cosine | $4.0 \times 10^{-5}$ |
| MMVP-VLM | CLIP-B16 | 2.0 | 0 | linear | $1.0 \times 10^{-5}$ |
| | SigLIP-B16 | 2.0 | 0.1 | cosine | $2.0 \times 10^{-5}$ |
| ColorSwap | CLIP-B16 | 2.3 | 0 | cosine | $4.0 \times 10^{-5}$ |
| | SigLIP-B16 | 1.0 | 0 | cosine | $4.0 \times 10^{-5}$ |
| | SigLIP-L16 | 2.5 | 0 | cosine | $4.0 \times 10^{-5}$ |

Table 5: Hyperparameters used for experiments in Section 4.3.

| Dataset | Model | $\tau_1$ | $\tau_T$ | Schedule | lr |
|---|---|---|---|---|---|
| Winoground | SmolVLM-256M-Instruct | 0.1 | 0 | linear | $4.0 \times 10^{-5}$ |
| MMVP-VLM | SmolVLM-256M-Instruct | 0.1 | 0 | linear | $6.0 \times 10^{-5}$ |
| ColorSwap | SmolVLM-256M-Instruct | 0.25 | 0 | linear | $6.0 \times 10^{-5}$ |

Table 6: Hyperparameters used for experiments in Section 4.4.

| **Variant** | **Model** | $\tau_1$ | $\tau_T$ | Schedule | lr |
|---|---|---|---|---|---|
| Replace Relation | SigLIP-B16 | 2.1 | 0 | cosine | $1.0 \times 10^{-5}$ |
| Swap Attribute | SigLIP-B16 | 1.8 | 0 | cosine | $1.0 \times 10^{-5}$ |
| Swap Object | SigLIP-B16 | 2.0 | 0 | cosine | $1.0 \times 10^{-5}$ |
| Add Attribute | SigLIP-B16 | 2.5 | 0 | cosine | $1.0 \times 10^{-5}$ |
| WhatsUp A (1×4) | CLIP-B32 | 0.55 | 0 | linear | $1.0 \times 10^{-5}$ |
| WhatsUp B (1×4) | CLIP-B32 | 0.80 | 0 | linear | $1.0 \times 10^{-5}$ |
| A-Left-Right (2×2) | CLIP-B32 | 0.25 | 0 | linear | $1.0 \times 10^{-5}$ |
| A-On-Under (2×2) | CLIP-B32 | 0.85 | 0 | linear | $1.0 \times 10^{-5}$ |
| B-Left-Right (2×2) | CLIP-B32 | 0.50 | 0 | cosine | $2.0 \times 10^{-5}$ |
| B-Front-Behind (2×2) | CLIP-B32 | 1.30 | 0 | cosine | $2.0 \times 10^{-5}$ |

Table 7: Hyperparameters used for experiments in Section 4.5. We adopt percentile-based thresholding: at iteration $t$, the top $1 - \tau_t$ fraction of pseudo-labels (ranked by similarity) is selected.

| Dataset | Model | $\tau_1$ | $\tau_T$ | Schedule | lr |
|---|---|---|---|---|---|
| Winoground | SigLIP-B16 | 0.50 | 0 | linear | $1.0 \times 10^{-5}$ |
| MMVP-VLM | SigLIP-B16 | 0.55 | 0 | linear | $2.0 \times 10^{-5}$ |
| ColorSwap | SigLIP-B16 | 0.50 | 0 | linear | $4.0 \times 10^{-5}$ |

## C.2 ANALYSES AND ABLATIONS

**Group matching provides strong supervision signals.** The key advantage of GroupMatch over GroupScore lies in its ability to leverage matching within local groups. To assess the benefits of matching and group structure, we examine the raw performance of CLIP-B16 and SigLIP-B16 under different evaluation metrics. In addition to GroupScore and GroupMatch, we consider (i) *global matching under Eq. (2)*, which performs matching but ignores group structure, and (ii) *individual matching within groups*, which preserves group structure but doesn't perform matching: it assign captions to images independently within the group. As shown in the left plot of Fig. 4, GroupMatch

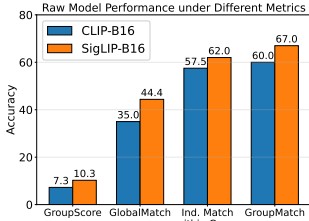 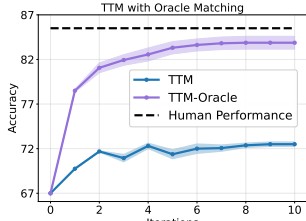 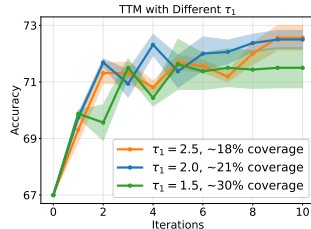

Figure 4: *Left:* Raw performance of CLIP-B16 and SigLIP-B16 on Winoground under different evaluation metrics. *Middle:* Skyline performance of TTM with oracle matching on Winoground with SigLIP-B16, illustrating the upper bound achievable by TTM. *Right:* Effect of the initial threshold $\tau_1$ on TTM performance, evaluated on Winoground with SigLIP-B16.

provides the strongest supervision signal among all metrics, making it most effective for guiding pseudo-labeling.

**Skyline performance with oracle matching.** To study the full potential of TTM, we evaluate an oracle variant that incorporates pseudo-labels into $\mathcal{S}_t$ if and only if they are correct (i.e., with oracle access). As shown in the middle plot of Fig. 4, this oracle variant enables TTM to bootstrap more aggressively, approaching human-level performance on Winoground. This suggests that improving pseudo-label quality—potentially through the incorporation of external supervision—could further enhance the effectiveness of TTM.

**Threshold selection for TTM.** As discussed in Section 3.2, we adopt a decaying threshold schedule that begins with high-quality pseudo-labels and gradually expands coverage. In our experiments, the final threshold $\tau_T$ is set to either $0$ (full coverage) or $0.1$ (typically covering more than 90% of the data). The optimal initial threshold $\tau_1$ is more dataset- and model-dependent. If a training set or hold-out split is available, $\tau_1$ can be selected based on matching results on that data (e.g., see the left and middle plots of Fig. 2). Otherwise, we find it effective to set $\tau_1$ such that roughly 15%–30% of the groups are matched initially. The right plot of Fig. 4 shows TTM results on Winoground with SigLIP-B16 for $\tau_1 \in \{2.5, 2, 1.5\}$, corresponding to roughly $\{18\%, 21\%, 30\%\}$ initial coverage. While performance varies slightly across these choices, all yield consistent gains, highlighting that TTM robustly improves model performance at test time. For the global matching variant, we find it effective to set $\tau_1$ such that about 50% of the data are pseudo-labeled initially. See Appendix C.1.2 for the complete threshold setups used in our experiments.

## C.3 COMPLETE RESULTS FROM SECTION 4.4

We present complete empirical results for Fig. 3 below in Tables 8 and 9. Following Li et al. (2025), we further convert the WhatsUp datasets into four directional variants with $2 \times 2$ group structures and present results in Table 10: Algorithm 1 again yields significant improvements—**up to 135.1% relative gains and 95.5% relative error reduction**—on top of SimpleMatch. Together, these results demonstrate that TTM is broadly effective across both $k \times k$ and $1 \times k$ settings, even in cases where evaluation metrics themselves cannot induce gains.

Table 8: Performance on SugarCrepe datasets ($1 \times 2$ groups) without metric-induced boosts: for $1 \times k$ groups, GroupScore and GroupMatch coincide. Raw SigLIP-B16 performance is reported under GroupScore, and TTM corresponds to the performance of Algorithm 1. We report absolute gains ($\Delta$), relative gains, and relative error reductions of TTM over the raw model performance.

| Datasets | SigLIP-B16 | TTM | $\Delta$ | Error Reduction |
|---|---|---|---|---|
| Replace Relation | 70.48 | **76.23** $\pm$ 0.51 | **+5.8** (8.2% ↑) | **19.5%** ↓ |
| Swap Attribute | 71.47 | **77.36** $\pm$ 0.71 | **+5.9** (8.2% ↑) | **20.6%** ↓ |
| Swap Object | 60.41 | **66.12** $\pm$ 2.06 | **+5.7** (9.5% ↑) | **14.4%** ↓ |
| Add Attribute | 83.67 | **88.95** $\pm$ 0.83 | **+5.3** (6.3% ↑) | **32.3%** ↓ |

Table 9: Performance on WhatsUp A/B datasets ($1 \times 4$ groups) without metric-induced boosts: for $1 \times k$ groups, GroupScore and GroupMatch coincide. Raw CLIP-B32 performance is reported under GroupScore, and TTM corresponds to the performance of Algorithm 1. We report absolute gains ($\Delta$), relative gains, and relative error reductions of TTM over the raw model performance.

| Datasets | CLIP-B32 | TTM | $\Delta$ | Error Reduction |
|---|---|---|---|---|
| WhatsUp A | 30.58 | $\mathbf{56.8}_{\pm 1.84}$ | $+26.2$ $\mathbf{(85.7\% \uparrow)}$ | $\mathbf{37.7\% \downarrow}$ |
| WhatsUp B | 30.88 | $\mathbf{49.94}_{\pm 2.58}$ | $+19.1$ $\mathbf{(61.7\% \uparrow)}$ | $\mathbf{27.6\% \downarrow}$ |

Table 10: Performance on WhatsUp $2 \times 2$ directional variants: LR: left-right, OU: on-under; FB: front-behind. Raw CLIP-B32 performance is reported under GroupScore. SimpleMatch corresponds to the performance under GroupMatch (Section 3.1), and TTM corresponds to the performance of Algorithm 1. We report absolute gains ($\Delta$), relative gains, and relative error reductions of TTM over SimpleMatch.

| Datasets | CLIP-B32 | SimpleMatch | TTM | $\Delta$ | Error Reduction |
|---|---|---|---|---|---|
| A-LR | 0 | 40.78 | $\mathbf{95.87}_{\pm 4.42}$ | $+55.1$ $\mathbf{(135.1\% \uparrow)}$ | $\mathbf{93.0\% \downarrow}$ |
| A-OU | 3.88 | 78.64 | $\mathbf{99.03}_{\pm 0}$ | $+20.4$ $\mathbf{(25.9\% \uparrow)}$ | $\mathbf{95.5\% \downarrow}$ |
| B-LR | 0 | 55.88 | $\mathbf{82.84}_{\pm 0.49}$ | $+27.0$ $\mathbf{(48.2\% \uparrow)}$ | $\mathbf{61.1\% \downarrow}$ |
| B-FB | 0 | 47.06 | $\mathbf{66.67}_{\pm 1.30}$ | $+19.6$ $\mathbf{(41.7\% \uparrow)}$ | $\mathbf{37.0\% \downarrow}$ |

## C.4 RUNTIME ANALYSIS AND EFFECTIVENESS OF TTM UNDER A SMALL NUMBER OF ITERATIONS

**Runtime analysis.** The runtime of TTM scales as $O(T \cdot C_{\text{ft}})$, where $T$ is the number of iterations and $C_{\text{ft}}$ denotes the model finetuning cost. While TTM also requires selecting the maximum-similarity matching within each group, this cost is dominated by the finetuning cost. Specifically:

- The per-group matching cost is $P(k) = O(k^3)$ for $k \times k$ groups (via the Hungarian algorithm) or $P(k) = O(k)$ for $1 \times k$ groups. Since compositional benchmarks use very small group sizes ($k = 2$ for $k \times k$ groups or $k = 4$ for $1 \times k$ groups), we can safely treat $P(k) = O(1)$.

- Matching over all $n$ groups therefore costs $O(n)$, which is dominated by the model finetuning cost $C_{\text{ft}}$.

Thus, the overall runtime is dominated by finetuning and scales as $O(T \cdot C_{\text{ft}})$. In practice, the finetuning cost $C_{\text{ft}}$ can be further reduced via efficient finetuning techniques (Hu et al., 2022).

**Effectiveness of TTM under A small number of iterations.** While we use $T = 10$ in our main experiments with contrastive vision-language models, we also evaluate TTM with $T = 3$ and $T = 5$. As shown Table 11, TTM continues to yield substantial gains even with only $T = 3$ or $T = 5$ iterations.

## D THE USE OF LARGE LANGUAGE MODELS (LLMS)

LLMs were used to polish the writing of this paper.

Table 11: Performance on Winoground, MMVP-VLM, and ColorSwap under varying number of iterations ($T$). Raw model performance is reported under GroupScore. SimpleMatch corresponds to the performance under GroupMatch (Section 3.1), and TTM corresponds to the performance of Algorithm 1. We report absolute gains ($\Delta$), relative gains, and relative error reductions of TTM over SimpleMatch.

| Dataset (Iterations) | Raw | SimpleMatch | TTM | $\Delta$ | Error Red. |
|---|---|---|---|---|---|
| **Winoground** | | | | | |
| SigLIP-B16 ($T = 3$) | 10.25 | 67.00 | $\mathbf{71.50}_{\pm 1.19}$ | $+\mathbf{4.5}\,(\mathbf{6.7\%}\uparrow)$ | $\mathbf{13.6\%}\downarrow$ |
| SigLIP-B16 ($T = 5$) | 10.25 | 67.00 | $\mathbf{71.88}_{\pm 1.48}$ | $+\mathbf{4.9}\,(\mathbf{7.3\%}\uparrow)$ | $\mathbf{14.8\%}\downarrow$ |
| SigLIP-B16 ($T = 10$) | 10.25 | 67.00 | $\mathbf{72.50}_{\pm 0.64}$ | $+\mathbf{5.5}\,(\mathbf{8.2\%}\uparrow)$ | $\mathbf{16.7\%}\downarrow$ |
| **MMVP-VLM** | | | | | |
| SigLIP-B16 ($T = 3$) | 22.96 | 81.48 | $\mathbf{85.19}_{\pm 0.91}$ | $+\mathbf{3.7}\,(\mathbf{4.6\%}\uparrow)$ | $\mathbf{20.0\%}\downarrow$ |
| SigLIP-B16 ($T = 5$) | 22.96 | 81.48 | $\mathbf{87.04}_{\pm 1.11}$ | $+\mathbf{5.6}\,(\mathbf{6.8\%}\uparrow)$ | $\mathbf{30.0\%}\downarrow$ |
| SigLIP-B16 ($T = 10$) | 22.96 | 81.48 | $\mathbf{89.44}_{\pm 0.96}$ | $+\mathbf{8.0}\,(\mathbf{9.8\%}\uparrow)$ | $\mathbf{43.0\%}\downarrow$ |
| **ColorSwap** | | | | | |
| SigLIP-B16 ($T = 3$) | 30.33 | 88.00 | $\mathbf{93.58}_{\pm 1.01}$ | $+\mathbf{5.6}\,(\mathbf{6.3\%}\uparrow)$ | $\mathbf{46.5\%}\downarrow$ |
| SigLIP-B16 ($T = 5$) | 30.33 | 88.00 | $\mathbf{93.58}_{\pm 0.86}$ | $+\mathbf{5.6}\,(\mathbf{6.3\%}\uparrow)$ | $\mathbf{46.5\%}\downarrow$ |
| SigLIP-B16 ($T = 10$) | 30.33 | 88.00 | $\mathbf{94.25}_{\pm 0.43}$ | $+\mathbf{6.3}\,(\mathbf{7.1\%}\uparrow)$ | $\mathbf{52.1\%}\downarrow$ |

