# OpenReview forum: "Test-Time Matching: Unlocking Compositional Reasoning in Multimodal Models"
_ICLR.cc/2026/Conference — ICLR 2026 Poster_

### Official Review · Reviewer_NrUV · 2025-10-23

**Soundness:** 3
**Presentation:** 3
**Contribution:** 2
**Rating:** 4
**Confidence:** 3

**Summary:**

This paper proposes Test-Time Matching (TTM), a method that enhances compositional reasoning in multimodal models by redefining evaluation metrics (GroupMatch) and applying unsupervised test-time adaptation. While the idea of leveraging matching structures is interesting, the improvements mainly come from metric relaxation rather than genuine reasoning gains. The proposed TTM is conceptually similar to prior test-time adaptation methods, incurs high computational cost, and shows limited practicality beyond small benchmarks.

**Strengths:**

1. The paper addresses an important problem in multimodal compositional reasoning and highlights biases in existing evaluation metrics.

2. The proposed GroupMatch metric and Test-Time Matching (TTM) framework offer an interesting perspective on improving model performance without additional supervision.

**Weaknesses:**

1. The paper mainly improves evaluation metrics (GroupMatch) rather than actual model reasoning ability. The performance gains may stem from relaxed scoring rather than genuine compositional understanding.

2. The Test-Time Matching (TTM) algorithm is conceptually similar to existing test-time adaptation methods (e.g., TENT, AdaContrast). The contribution appears incremental rather than fundamentally new.

3. TTM requires iterative matching and fine-tuning during inference, which is computationally expensive and unsuitable for real-world deployment or online systems.

4. The reported improvements are mostly on small datasets (e.g., Winoground). It is unclear whether the method generalizes to larger or more realistic multimodal tasks.

**Questions:**

1. Since much of the reported improvement comes from the new GroupMatch metric, does the proposed method still yield significant gains under conventional metrics? The dependence on the new metric raises concerns about fair comparison.

2. TTM requires iterative matching and fine-tuning during inference. How does this impact efficiency compared to standard zero-shot evaluation? A runtime analysis would be helpful.

---

> ### Author Response · Authors · 2025-11-21
> **Author response (1/3)**
>
> Dear Reviewer,
>
> Thank you for taking the time to review our paper. We believe several key points of our work were misunderstood, and we appreciate the opportunity to clarify these points. Below, we first provide the key clarifications and then respond to your concerns individually. We would appreciate it if you could reconsider your assessment in light of these explanations.
>
> ---
>
> ### **Our SimpleMatch and TTM results remain valid under the original GroupScore metric.**
>
> A central misunderstanding appears to be the belief that our GroupMatch metric inflates model performance. This is not the case.
>
> Whenever GroupMatch obtains a perfect score, the induced matching is—by definition—the ground-truth matching. Thus, by overfitting to that matching, **correctness under GroupMatch can be converted into correctness under GroupScore.** This is exactly the idea behind SimpleMatch (lines 183–192), and our empirical results confirm this behavior.
>
> Consequently, **all results (except Table 2) on SimpleMatch and TTM remain valid under the original GroupScore metric.** Table 2 is the only exception because we intentionally removed the local group structure to test the global version of our approach. If the local grouping is restored—as in GroupScore—the performance in Table 2 returns to the level of Table 1, which is valid under GroupScore.
>
> ### **GroupScore artificially deflates model performance.**
>
> This follows from two facts: (i) correctness under GroupMatch corresponds to the ground-truth matching and can be directly converted into correctness under GroupScore (operationalized via SimpleMatch), and (ii) GroupMatch is mathematically easier to satisfy (its constraints are strictly weaker than those of GroupScore; see Definitions and Propositions 1 and 2). Together, these imply that **GroupScore is the problematic metric that artificially deflates model performance**: it imposes unnecessary constraints and can miscount correct matchings as wrong. This artifact has led to the long-standing misconception that well-trained multimodal models perform “near random guessing” on compositional reasoning tasks—a counterintuitive conclusion that arises solely from the limitations of GroupScore.
>
> To better understand this, consider a $2 \times 2$ group with two captions and two images. *There are only 2 possible matchings*: either (i) match image 1 to caption 1 and image 2 to caption 2, or (ii) match image 1 to caption 2 and image 2 to caption 1. So, even under random guessing, the success probability should be $1/2$. However, GroupScore adds unnecessary constraints and artificially deflates the success probability to $1/6$ (see Proposition 1).
>
> **We introduce GroupMatch to fix this issue and to provide a more faithful measure of model capability**. Because GroupScore is the default metric in widely used and highly cited benchmarks (>500 citations), we view this correction as an important contribution of our work.
>
> Given the above clarifications, we next respond to your concerns and questions individually.
>
> ---
>
> **Responses to Weaknesses:**
>
> >**Weakness 1:** The paper mainly improves evaluation metrics (GroupMatch) rather than actual model reasoning ability. The performance gains may stem from relaxed scoring rather than genuine compositional understanding.
>
> **Response:** While GroupMatch is mathematically easier to satisfy, it doesn’t inflate model performance because **correctness under GroupMatch corresponds to the ground-truth matching and can be directly converted into correctness under GroupScore**. In contrast, GroupScore is the problematic metric that artificially deflates model performance. Please see the clarification at the beginning of our rebuttal for details.

---

> ### Author Response · Authors · 2025-11-21
> **Author response (2/3)**
>
> >**Weakness 2:** The Test-Time Matching (TTM) algorithm is conceptually similar to existing test-time adaptation methods (e.g., TENT, AdaContrast). The contribution appears incremental rather than fundamentally new.
>
> **Response:** While TTM shares similarities with existing test-time training/adaptation methods (e.g., TENT, AdaContrast, Sun et al., 2020; Akyürek et al., 2024) in the sense that all of them finetune models at test time, our contribution is not incremental for two reasons:
>
> **TTM introduces a new source of supervision.** TTM leverages **GroupMatch-based pseudo-labels**, which are only made possible because we propose GroupMatch to fix the long-standing issues in GroupScore. Please see the clarification at the beginning of our rebuttal for details.
>
> **The Decay schedule is key to TTM’s consistent improvements.** TTM (and any pseudo-label-based method) faces two types of pseudo-labeling errors: (i) *false-positive errors*—incorrect matchings mistakenly included in the training set, and (ii) *false-negative errors*—correctly predicted matchings that are excluded because the threshold is too strict. The **Decay schedule** of TTM is specifically designed to *balance these two types of errors*: It begins with a high threshold to ensure high-precision pseudo-labels (few false positives), and then gradually lowers the threshold to increase coverage and reduce false negatives as the model improves. This dynamic balance enabled by the Decay schedule is central to TTM’s effectiveness and explains why it delivers consistent and substantial gains across different models and 16 dataset variants. In contrast, standard schedules provide no or limited improvements (Fig. 2, right).
>
> Thank you for the pointers. We will add clarifications and further discussion of these points in the revision.
>
> >**Weakness 3:** TTM requires iterative matching and fine-tuning during inference, which is computationally expensive and unsuitable for real-world deployment or online systems.
>
> **Response:** Thank you for your comment. We first clarify that the runtime of TTM is **comparable to standard test-time training methods** in the literature (e.g., Sun et al., 2020; Akyürek et al., 2024); please see our response to Question #2 below for a detailed runtime discussion.
>
> More importantly, while finetuning incurs additional cost compared to zero-shot evaluation, **TTM provides substantial and consistent gains across different models and 16 dataset variants.** These improvements are impressive: **TTM enables a 0.2B SigLIP-B16 model to outperform GPT-4.1 on MMVP-VLM**; for $1 \times k$ groups where GroupMatch and GroupScore coincide (so SimpleMatch alone doesn’t improve performance), TTM still improves performance up to 85.7%. These results demonstrate that TTM can meaningfully enhance compositional reasoning in settings where accuracy is a priority.
>
> >**Weakness 4:** The reported improvements are mostly on small datasets (e.g., Winoground). It is unclear whether the method generalizes to larger or more realistic multimodal tasks.
>
> **Response:** Thank you for your comment. We first note that popular compositional reasoning benchmarks are relatively small because such datasets are difficult to curate at scale. Our experiments therefore follow standard practice in this literature.
>
> We expect TTM to benefit from larger datasets in practice since having more data typically leads to stronger self-supervision. However, even in the worst case, the argument below guarantees that TTM’s performance on a larger dataset remains consistent with its performance on smaller ones.
>
> **The worst-case argument.** A larger dataset can be viewed as a collection of smaller disjoint subsets. We can then apply TTM independently on each subset and average the results. Consequently, TTM’s performance on larger datasets remains consistent with its performance on smaller subsets. This shows that TTM is not inherently tied to small dataset sizes.

---

> ### Author Response · Authors · 2025-11-21
> **Author response (3/3)**
>
> **Responses to Questions:**
>
> >**Question 1:** Since much of the reported improvement comes from the new GroupMatch metric, does the proposed method still yield significant gains under conventional metrics? The dependence on the new metric raises concerns about fair comparison.
>
> **Response:** Thank you for the question. **Yes, our SimpleMatch and TTM results remain valid under the original GroupScore metric** since correctness under GroupMatch corresponds to the ground-truth matching and can be directly converted into correctness under GroupScore (operationalized via SimpleMatch). Please see the clarification at the beginning of our rebuttal for details.
>
> >**Question 2:** TTM requires iterative matching and fine-tuning during inference. How does this impact efficiency compared to standard zero-shot evaluation? A runtime analysis would be helpful.
>
> **Response:** Thank you for the question. The runtime of TTM scales as $O(T \cdot C_{ft})$, where $T$ is the number of iterations (can be as small as $3$; see results below) and $C_{ft}$ denotes the model finetuning cost. ​​A detailed runtime analysis is provided below.
>
> While TTM incurs additional cost compared to zero-shot evaluation—whose runtime largely scales with the number of test examples—TTM provides substantial performance gains: it consistently improves results across different models and 16 dataset variants. Remarkably, **TTM enables a 0.2B SigLIP-B16 model to outperform GPT-4.1 on MMVP-VLM**. These results show that TTM can meaningfully enhance compositional reasoning in settings where accuracy is a priority.
>
> **Additional results.** While we use $T=10$ in our main experiments, we also evaluate TTM with $T=3$ and $T=5$. As shown below, TTM continues to yield substantial gains even with only a few iterations.
>
> **TTM achieves strong performance even with $T=3$ and $5$ iterations.**
> | **Datasets (Iterations)**       | **GroupScore** | **SimpleMatch** | **TTM**              | **Δ over SimpleMatch (Relative)** | **Error Reduction** |
> |:--------------|:--------------:|:---------------:|:---------------------:|:---------------------------------:|:--------------:|
> | **Winoground ($T=10$)** | 10.25 | 67.00 | **72.50** ± 0.64 | **+5.5 (8.2% ↑)** | **16.7% ↓** |
> | **Winoground ($T=5$)**  | 10.25 | 67.00 | **71.88** ± 1.48 | **+4.9 (7.3% ↑)** | **14.8% ↓** |
> | **Winoground ($T=3$)**  | 10.25 | 67.00 | **71.5** ± 1.19 | **+4.5 (6.7% ↑)** | **13.6% ↓** |
> | **MMVP-VLM ($T=10$)**   | 22.96 | 81.48 | **89.44** ± 0.96 | **+8.0 (9.8% ↑)** | **43.0% ↓** |
> | **MMVP-VLM ($T=5$)**    | 22.96 | 81.48 | **87.04** ± 1.11 | **+5.6 (6.8% ↑)** | **30.0% ↓** |
> | **MMVP-VLM ($T=3$)**    | 22.96 | 81.48 | **85.19** ± 0.91 | **+3.7 (4.6% ↑)** | **20.0% ↓** |
> | **ColorSwap ($T=10$)**  | 30.33 | 88.00 | **94.25** ± 0.43 | **+6.3 (7.1% ↑)** | **52.1% ↓** |
> | **ColorSwap ($T=5$)**   | 30.33 | 88.00 | **93.58** ± 0.86 | **+5.6 (6.3% ↑)** | **46.5% ↓** |
> | **ColorSwap ($T=3$)**   | 30.33 | 88.00 | **93.58** ± 1.01 | **+5.6 (6.3% ↑)** | **46.5% ↓** |
>
> **Detailed runtime analysis.** The runtime of TTM scales as $O(T \cdot C_{ft})$, where $T$ is the number of iterations and $C_{ft}$ denotes the model finetuning cost. Although TTM also performs a matching step within each group, this overhead is negligible compared to finetuning. Specifically:
>
> - The per-group matching cost is $P(k) = O(k^3)$ for $k \times k$ groups (via the Hungarian algorithm) or $P(k) = O(k)$ for $1 \times k$ groups. Since compositional benchmarks use very small group sizes ($k=2$ for $k \times k$ groups or $k=4$ for $1 \times k$ groups), we can safely treat $P(k) = O(1)$.
> - Matching over all $n$ groups therefore costs $O(n)$, which is negligible relative to the finetuning cost $C_{ft}$.
>
> Thus, the overall runtime is dominated by finetuning and scales as $O(T \cdot C_{ft})$. In practice, the finetuning cost $C_{ft}$ can be further reduced via efficient finetuning techniques such as PEFT.

---

> > ### Comment · Reviewer_NrUV · 2025-11-28
> > **Response to Authors**
> >
> > I thank the authors for their detailed response. I particularly appreciate the clarification of the equivalence between GroupMatch and GroupScore, as well as the efficiency analysis. These points have addressed my concerns, and I have changed my view of this work to a positive one. However, please note that this is not my primary area of expertise; therefore, I will continue to follow the discussion and consider the other reviewers' comments.

---

> > > ### Author Response · Authors · 2025-11-28
> > >
> > > Thank you for your response and for sharing your updated perspective on our work. We are glad that our clarifications addressed your concerns. We will continue to engage in the discussion, and please feel free to let us know if any further questions arise.

---

### Official Review · Reviewer_4Q99 · 2025-10-26

**Soundness:** 2
**Presentation:** 3
**Contribution:** 2
**Rating:** 2
**Confidence:** 4

**Summary:**

This paper proposes a more relaxed and losser evaluation metric, GroupMatch, for multimodal compositional reasoning and an iterative test-time finetuning method (TTM). Instead of modifying model architectures, the authors show that changing the metric from GroupScore to GroupMatch can boost the performance of existing models. Also, the author claim that TTM further boosts performance through the pseudo-label–based training of model parameters at test time.

**For me, I highly believe this work is lower than the bar of ICLR.** The reasons are one-by-one listed in weaknesses.

**Strengths:**

1. The authors propose a simple but effective TTM method with impressive empirical gains.
2. The authors claim that the results achieved by the method surpass human performance on Winoground.

**Weaknesses:**

1. **Lack of clear rationale and rigorousness.** The motivation behind using a looser metric is not fully clear. The authors should give more explaniation for the connection between the relaxed metric GroupMatch and the real reasoning ability of models. Does GroupMatch truly measure reasoning ability, or just use a looser rule to make results look good? I still believe that the standrad strict metric is more rigorous than the GroupMatch desinged in this paer.

2. **Risk of inflation.** Because GroupMatch is easier to satisfy, models may appear much stronger without any actual improvement in reasoning. This could distort comparisons across methods or benchmarks. I indeed do not believe using a looser evaluation metric is absolutely better than the use of a strict metric.

3. **Lack of theoretical grounding.** The paper claims that GroupMatch “reveals hidden capabilities,” but provides limited formal justification or analysis on why this metric better reflects reasoning.

4. **Lack of theoretical grounding.** The paper could provide more explicit analysis of the runtime/efficiency implications of TTM, especially for large-scale deployments.

5. **Limited robustness analysis.** There is no clear evaluation of how sensitive GroupMatch is to noise, adversarial perturbations, or different group sizes. If the metric is fragile, its reliability as an evaluation tool is questionable.

6. **Unconvincing results.** Since TTM explicitly uses the test data for adaptation, and GroupMatch relaxes the matching requirement. So, the authors should report the performance improvment brought by TTM using the original stric GroupScore, which makes the restults more solid.

7. **Poor practical applicability.** TTM need to finetue the model' parameters on test data, which has worse practical applicability in real-world scenarios, e.g., test data may be scarce and models may be very large (such as GPT-4,5, and Gemini).

**Questions:**

1. Does the method generalize beyond image-text tasks to other modalities (e.g., audio-text, video-text)?

2. Does GroupMatch truly measure reasoning ability, or just alignment under a softer rule?

3. Can models cheat the metric GroupMatch without improving their underlying reasoning as it is more looser and relaxed?

---

> ### Author Response · Authors · 2025-11-21
> **Author response (1/3)**
>
> Dear Reviewer,
>
> Thank you for taking the time to review our paper. We believe several key points of our work were misunderstood, which directly shaped many of the concerns listed in the weaknesses. We appreciate the opportunity to clarify these points. Below, we first provide the key clarifications and then respond to your concerns individually. We would appreciate it if you could reconsider your assessment in light of these explanations.
>
> ---
>
> ### **Our SimpleMatch and TTM results remain valid under the original GroupScore metric.**
>
> A central misunderstanding appears to be the belief that our GroupMatch metric inflates model performance. This is not the case.
>
> Whenever GroupMatch obtains a perfect score, the induced matching is—by definition—the ground-truth matching. Thus, by overfitting to that matching, **correctness under GroupMatch can be converted into correctness under GroupScore.** This is exactly the idea behind SimpleMatch (lines 183–192), and our empirical results confirm this behavior.
>
> Consequently, **all results (except Table 2) on SimpleMatch and TTM remain valid under the original GroupScore metric.** Table 2 is the only exception because we intentionally removed the local group structure to test the global version of our approach. If the local grouping is restored—as in GroupScore—the performance in Table 2 returns to the level of Table 1, which is valid under GroupScore.
>
> ### **GroupScore artificially deflates model performance.**
>
> This follows from two facts: (i) correctness under GroupMatch corresponds to the ground-truth matching and can be directly converted into correctness under GroupScore (operationalized via SimpleMatch), and (ii) GroupMatch is mathematically easier to satisfy (its constraints are strictly weaker than those of GroupScore; see Definitions and Propositions 1 and 2). Together, these imply that **GroupScore is the problematic metric that artificially deflates model performance**: it imposes unnecessary constraints and can miscount correct matchings as wrong. This artifact has led to the long-standing misconception that well-trained multimodal models perform “near random guessing” on compositional reasoning tasks—a counterintuitive conclusion that arises solely from the limitations of GroupScore.
>
> To better understand this, consider a $2 \times 2$ group with two captions and two images. *There are only 2 possible matchings*: either (i) match image 1 to caption 1 and image 2 to caption 2, or (ii) match image 1 to caption 2 and image 2 to caption 1. So, even under random guessing, the success probability should be $1/2$. However, GroupScore adds unnecessary constraints and artificially deflates the success probability to $1/6$ (see Proposition 1).
>
> **We introduce GroupMatch to fix this issue and to provide a more faithful measure of model capability**. Because GroupScore is the default metric in widely used and highly cited benchmarks (>500 citations), we view this correction as an important contribution of our work.
>
> Given the above clarifications, we next respond to your concerns and questions individually.
>
> ---
>
> **Responses to Weaknesses:**
>
> >**Weakness 1:** Lack of clear rationale and rigorousness. The motivation behind using a looser metric is not fully clear. The authors should give more explaniation for the connection between the relaxed metric GroupMatch and the real reasoning ability of models. Does GroupMatch truly measure reasoning ability, or just use a looser rule to make results look good? I still believe that the standrad strict metric is more rigorous than the GroupMatch desinged in this paer.
>
> **Response:** We introduce GroupMatch to fix issues with the widely used GroupScore metric, which artificially deflates model performance. As clarified before, **correctness under GroupMatch corresponds to the ground-truth matching and can be directly converted into correctness under GroupScore**, so GroupMatch does not inflate model performance. In contrast, **GroupScore is the problematic metric that artificially deflates model performance**: it imposes unnecessary constraints and can miscount correct matchings as wrong. GroupScore has led to the long-standing misconception that well-trained multimodal models perform “near random guessing” on compositional reasoning tasks. We introduce GroupMatch to fix this issue and to provide a more faithful measure of model capability. Because GroupScore is the default metric in widely used and highly cited benchmarks (>500 citations), we view this correction as an important contribution of our work.

---

> ### Author Response · Authors · 2025-11-21
> **Author response (2/3)**
>
> >**Weakness 2:** Risk of inflation. Because GroupMatch is easier to satisfy, models may appear much stronger without any actual improvement in reasoning. This could distort comparisons across methods or benchmarks. I indeed do not believe using a looser evaluation metric is absolutely better than the use of a strict metric.
>
> **Response:** As clarified before, while GroupMatch is mathematically easier to satisfy, it doesn’t inflate performance because **correctness under GroupMatch corresponds to the ground-truth matching and can be directly converted into correctness under GroupScore**. In contrast, GroupScore is the problematic metric that artificially deflates model performance. Please see the clarification at the beginning of our rebuttal for details.
>
> >**Weakness 3:** Lack of theoretical grounding. The paper claims that GroupMatch “reveals hidden capabilities,” but provides limited formal justification or analysis on why this metric better reflects reasoning.
>
> **Response:** We provide the formal justification in Section 3.1. GroupMatch reveals hidden capabilities because **GroupScore is the problematic metric that artificially deflates model performance**. This follows from two facts: (i) correctness under GroupMatch corresponds to the ground-truth matching and can be directly converted into correctness under GroupScore (operationalized via SimpleMatch), and (ii) GroupMatch is mathematically easier to satisfy (its constraints are strictly weaker than those of GroupScore; see Definitions and Propositions 1 and 2). Together, these imply that GroupScore is the problematic metric that artificially deflates model performance: it imposes unnecessary constraints and can miscount correct matchings as wrong. We introduce GroupMatch to fix this issue and to provide a more faithful measure of model capability.
>
> >**Weakness 4:** Lack of theoretical grounding. The paper could provide more explicit analysis of the runtime/efficiency implications of TTM, especially for large-scale deployments.
>
> **Response:** Thank you for your suggestion. The runtime of TTM scales as $O(T \cdot C_{ft})$, where $T$ is the number of iterations and $C_{ft}$ denotes the model finetuning cost (see detailed runtime analysis below). Importantly, TTM achieves strong performance even with very small $T$ (e.g., 3 or 5; see results below), which keeps the runtime **comparable to standard test-time training methods** in the literature (e.g., Sun et al., 2020; Akyürek et al., 2024).
>
> **Additional results.** While we use $T=10$ in our main experiments, we also evaluate TTM with $T=3$ and $T=5$. As shown below, TTM continues to yield substantial gains even with only a few iterations.
>
> **TTM achieves strong performance even with $T=3$ and $5$ iterations.**
> | **Datasets (Iterations)**       | **GroupScore** | **SimpleMatch** | **TTM**              | **Δ over SimpleMatch (Relative)** | **Error Reduction** |
> |:--------------|:--------------:|:---------------:|:---------------------:|:---------------------------------:|:--------------:|
> | **Winoground ($T=10$)** | 10.25 | 67.00 | **72.50** ± 0.64 | **+5.5 (8.2% ↑)** | **16.7% ↓** |
> | **Winoground ($T=5$)**  | 10.25 | 67.00 | **71.88** ± 1.48 | **+4.9 (7.3% ↑)** | **14.8% ↓** |
> | **Winoground ($T=3$)**  | 10.25 | 67.00 | **71.5** ± 1.19 | **+4.5 (6.7% ↑)** | **13.6% ↓** |
> | **MMVP-VLM ($T=10$)**   | 22.96 | 81.48 | **89.44** ± 0.96 | **+8.0 (9.8% ↑)** | **43.0% ↓** |
> | **MMVP-VLM ($T=5$)**    | 22.96 | 81.48 | **87.04** ± 1.11 | **+5.6 (6.8% ↑)** | **30.0% ↓** |
> | **MMVP-VLM ($T=3$)**    | 22.96 | 81.48 | **85.19** ± 0.91 | **+3.7 (4.6% ↑)** | **20.0% ↓** |
> | **ColorSwap ($T=10$)**  | 30.33 | 88.00 | **94.25** ± 0.43 | **+6.3 (7.1% ↑)** | **52.1% ↓** |
> | **ColorSwap ($T=5$)**   | 30.33 | 88.00 | **93.58** ± 0.86 | **+5.6 (6.3% ↑)** | **46.5% ↓** |
> | **ColorSwap ($T=3$)**   | 30.33 | 88.00 | **93.58** ± 1.01 | **+5.6 (6.3% ↑)** | **46.5% ↓** |
>
> **Detailed runtime analysis.** The runtime of TTM scales as $O(T \cdot C_{ft})$, where $T$ is the number of iterations and $C_{ft}$ denotes the model finetuning cost. Although TTM also performs a matching step within each group, this overhead is negligible compared to finetuning. Specifically:
>
> - The per-group matching cost is $P(k) = O(k^3)$ for $k \times k$ groups (via the Hungarian algorithm) or $P(k) = O(k)$ for $1 \times k$ groups. Since compositional benchmarks use very small group sizes ($k=2$ for $k \times k$ groups or $k=4$ for $1 \times k$ groups), we can safely treat $P(k) = O(1)$.
> - Matching over all $n$ groups therefore costs $O(n)$, which is negligible relative to the finetuning cost $C_{ft}$.
>
> Thus, the overall runtime is dominated by finetuning and scales as $O(T \cdot C_{ft})$. In practice, the finetuning cost $C_{ft}$ can be further reduced via efficient finetuning techniques such as PEFT.

---

> ### Author Response · Authors · 2025-11-21
> **Author response (3/3)**
>
> >**Weakness 5:** Limited robustness analysis. There is no clear evaluation of how sensitive GroupMatch is to noise, adversarial perturbations, or different group sizes. If the metric is fragile, its reliability as an evaluation tool is questionable.
>
> **Response:** Thank you for your comments. We first note that both GroupMatch and GroupScore operate on the same (model-derived) similarity matrix, so any noise or adversarial perturbations affect both GroupMatch and GroupScore. The important point is that **GroupMatch is strictly more robust as an evaluation metric** because, mathematically, its success region is a **superset** of that of GroupScore (see constraints in their definitions). This guarantees that, under any perturbation, **GroupMatch can never have worse performance than GroupScore.** We will clarify this in the revision.
>
> Both GroupScore and GroupMatch can be extended to general rectangular groups of size $m \times k$ (with $m < k$). In this case, the same structure holds: the success region of GroupMatch is a superset of the success region of GroupScore, so sGroupMatch continues to be the more robust metric.
>
> >**Weakness 6:** Unconvincing results. Since TTM explicitly uses the test data for adaptation, and GroupMatch relaxes the matching requirement. So, the authors should report the performance improvment brought by TTM using the original stric GroupScore, which makes the restults more solid.
>
> **Response:** As clarified before, our TTM results **remain valid under the original metric GroupScore** since correctness under GroupMatch corresponds to the ground-truth matching and can be directly converted into correctness under GroupScore. Please see the clarification at the beginning of our rebuttal for details.
>
> >**Weakness 7:** Poor practical applicability. TTM need to finetue the model' parameters on test data, which has worse practical applicability in real-world scenarios, e.g., test data may be scarce and models may be very large (such as GPT-4,5, and Gemini).
>
> **Response:** We first clarify that the runtime of TTM is **comparable to standard test-time training methods** (e.g., Sun et al., 2020; Akyürek et al., 2024); please see our response to Weakness #4 above for a detailed runtime analysis. Also, TTM is a form of test-time training (line 267), which—*by design*—updates model parameters on test inputs, as any other test-time training algorithm does.
>
> More importantly, while finetuning incurs additional cost compared to zero-shot evaluation, **TTM provides substantial and consistent gains across different models and 16 dataset variants.** As acknowledged in your review, these improvements are impressive: **TTM enables a 0.2B SigLIP-B16 model to outperform GPT-4.1 on MMVP-VLM**; for $1 \times k$ groups where GroupMatch and GroupScore coincide (so SimpleMatch alone doesn’t improve performance), TTM still improves performance up to 85.7%. These results demonstrate that TTM can meaningfully enhance compositional reasoning in settings where accuracy is a priority.
>
> **Responses to Questions:**
>
> >**Question 1:** Does the method generalize beyond image-text tasks to other modalities (e.g., audio-text, video-text)?
>
> **Response:** Yes. What matters for TTM is the presence of group structures, not the specific modalities involved.  We focus on image-text tasks simply because they are the standard benchmarks for compositional reasoning.
>
> >**Question 2:** Does GroupMatch truly measure reasoning ability, or just alignment under a softer rule?
>
> **Response:** While GroupMatch is mathematically easier to satisfy, it doesn’t inflate model performance because **correctness under GroupMatch can be directly converted into correctness under GroupScore**. GroupMatch measures the model’s actual reasoning ability—whether it found the correct matching—without the unnecessary constraints that cause GroupScore to miscount correct matchings as wrong. Please see the clarification at the beginning of our rebuttal for details.
>
> >**Question 3:** Can models cheat the metric GroupMatch without improving their underlying reasoning as it is more looser and relaxed?
>
> **Response:** GroupMatch cannot be “cheated” and does not inflate model performance because correctness under GroupMatch corresponds to the ground-truth matching and can be directly converted into correctness under GroupScore. GroupMatch does not mark incorrect matchings as correct; it simply removes unnecessary constraints that cause GroupScore to miscount correct matchings as wrong. Please see the clarification at the beginning of our rebuttal for details.

---

### Official Review · Reviewer_smg6 · 2025-10-28

**Soundness:** 3
**Presentation:** 2
**Contribution:** 3
**Rating:** 6
**Confidence:** 2

**Summary:**

This paper revisits the problem of compositional reasoning in multimodal models, which are often reported to perform near random chance on benchmarks such as Winoground. The authors first identify that existing evaluation metrics (notably GroupScore) systematically underestimate model capability. They propose a new metric, the Group Matching Score (GroupMatch), which evaluates the best overall image–caption matching within each group. This new metric reveals hidden competence in both contrastive VLMs (e.g., CLIP, SigLIP) and MLLMs (e.g., GPT-4.1), enabling GPT-4.1 to surpass estimated human performance on Winoground. Building on this, they introduce Test-Time Matching (TTM), an iterative, supervision-free self-training algorithm that converts confident model-induced matchings into pseudo-labels, progressively bootstrapping model performance during inference. TTM yields further non-trivial improvements—e.g., SigLIP-B16 surpasses GPT-4.1 on MMVP-VLM—achieving new state-of-the-art results across 16 benchmark variants, including datasets with and without group structures. In summary, the paper’s key contributions are: (1) redefining evaluation through GroupMatch to uncover hidden compositional ability; (2) proposing TTM for test-time self-bootstrapping without external supervision; and (3) demonstrating broad applicability and new SOTAs across both grouped and non-grouped multimodal reasoning benchmarks

**Strengths:**

The paper reframe multimodal compositional reasoning evaluation. By identifying that existing metrics systematically underestimate model ability, it introduces the GroupMatch score as a new evaluation paradigm that better reflects true multimodal alignment. Moreover, the proposed Test-Time Matching (TTM) algorithm creatively extends test-time training and self-training principles to multimodal evaluation without external supervision. This combination of metric reformulation and unsupervised adaptation constitutes a distinctly original contribution in the field. The derivation of probabilistic properties of existing and proposed metrics offers analytical clarity on why previous benchmarks yield misleadingly low scores. The empirical evaluations are comprehensive—covering 16 datasets and multiple model families (CLIP, SigLIP, GPT-4.1)—and are backed by detailed ablations, sensitivity analyses, and comparisons. The results are consistent and reproducible, indicating high experimental quality.

**Weaknesses:**

1. The SimpleMatch and TTM procedures explicitly overfit to test data using pseudo-labels derived from the model’s own predictions. Although the paper positions this as “test-time adaptation,” it effectively tunes on test distributions, which may blur the boundary between evaluation and training. The authors should clarify the implications of this for benchmark comparability and fairness, and discuss possible countermeasures such as validation-based stopping criteria or out-of-distribution robustness checks.

2. Although the paper covers 16 dataset variants, most are derived from similar group-based compositional reasoning benchmarks (Winoground, MMVP-VLM, Colorswap, SugarCrepe, Whatsup). It remains unclear whether TTM generalizes to broader multimodal tasks such as retrieval, captioning, or visual reasoning benchmarks like ARO or GQA. Testing beyond compositional group setups would strengthen claims about general applicability.

3. While the paper presents sensitivity studies on threshold decay, it does not deeply analyze the types of errors propagated during pseudo-labeling or how inaccurate matchings affect subsequent iterations. An empirical study on pseudo-label noise, or visualization of incorrect matchings and their impact, could make the bootstrapping dynamics more interpretable and trustworthy.

4. The iterative self-training procedure (ten iterations, multiple epochs each) likely increases inference and adaptation time significantly. The paper would benefit from reporting computational overheads, memory costs, and convergence behavior, as well as exploring lightweight or one-shot alternatives to make TTM practical for real-world applications.

5. The paper would benefit from a more formal characterization of when and why TTM converges, or how pseudo-label noise affects performance over iterations. Connections to existing theories in semi-supervised or test-time training (e.g., Sun et al., 2020; Gandelsman et al., 2022; Akyürek et al., 2024) could strengthen the methodological depth.

**Questions:**

Please see weaknesses.

---

> ### Author Response · Authors · 2025-11-21
> **Author response (1/3)**
>
> Thank you for taking the time to review our paper and for recognizing our contributions in fixing evaluation metrics and substantially improving model performance. Below, we provide detailed responses to your remaining concerns. If our clarifications address these points, we would appreciate it if you could consider updating your assessment.
>
> **Responses to Weaknesses:**
>
> >**Weakness 1:** The SimpleMatch and TTM procedures explicitly overfit to test data using pseudo-labels derived from the model’s own predictions. Although the paper positions this as “test-time adaptation,” it effectively tunes on test distributions, which may blur the boundary between evaluation and training. The authors should clarify the implications of this for benchmark comparability and fairness, and discuss possible countermeasures such as validation-based stopping criteria or out-of-distribution robustness checks.
>
> **Response:** Thank you for your comment. TTM is a form of test-time training (line 267), which—*by design*—updates model parameters on test inputs without accessing ground-truth labels. This setting is standard in the test-time training literature (e.g., Sun et al., 2020, Akyürek et al., 2024), and the evaluation is fair within this protocol: **TTM, like any other test-time training method, uses only the test inputs and no ground-truth labels**.
>
> TTM stops after a fixed number of iterations (e.g., 10), which works well in our experiments and consistently improves performance across 16 dataset variants. TTM is also robust to OOD tasks: as shown in Fig. 3 (middle), it improves raw performance from 30.6 to 56.8, turning these challenging OOD cases into tractable ones.
>
> >**Weakness 2:** Although the paper covers 16 dataset variants, most are derived from similar group-based compositional reasoning benchmarks (Winoground, MMVP-VLM, Colorswap, SugarCrepe, Whatsup). It remains unclear whether TTM generalizes to broader multimodal tasks such as retrieval, captioning, or visual reasoning benchmarks like ARO or GQA. Testing beyond compositional group setups would strengthen claims about general applicability.
>
> **Response:** Thank you for your comment. Our paper specifically focuses on compositional reasoning tasks, as reflected in the title and scope of the submission. Within this problem setting, our evaluation is comprehensive: as you also noted, we cover 16 dataset variants spanning multiple group configurations. Extending TTM to general multimodal tasks may require additional adaptations due to structural differences among tasks, which is outside the scope of the current submission. We therefore view such extensions as promising directions for future work.
>
> >**Weakness 3:** While the paper presents sensitivity studies on threshold decay, it does not deeply analyze the types of errors propagated during pseudo-labeling or how inaccurate matchings affect subsequent iterations. An empirical study on pseudo-label noise, or visualization of incorrect matchings and their impact, could make the bootstrapping dynamics more interpretable and trustworthy.
>
> **Response:** Thank you for your suggestion. We would like to clarify that our paper *does* empirically examine how pseudo-labeling errors affect TTM, as shown in Fig. 2 (right). Below we make this connection more explicit.
>
> TTM is influenced by two types of pseudo-labeling errors: (i) **false-positive errors**—incorrect matchings mistakenly included in the training set, and (ii) **false-negative errors**—correctly predicted matchings that are excluded because the threshold is too strict.
>
> Fig. 2 (right) illustrates how these errors propagate:
> - The **Ascend schedule** begins with a low threshold, which admits many false-positive matchings in early iterations. These noisy pseudo-labels degrade performance and prevent further improvement.
> - The **Constant schedule** avoids early false positives (via a high threshold), but its fixed threshold causes many false-negative errors in later iterations—correctly predicted matchings remain excluded—leading to limited improvement and early plateauing.
>
> Our **Decay schedule** is specifically designed to balance these two types of errors: It begins with a high threshold to ensure high-precision pseudo-labels (few false positives), and then gradually lowers the threshold to increase coverage and reduce false negatives as the model improves.
>
> This dynamic balance enabled by the Decay schedule is central to TTM’s effectiveness and explains why it delivers consistent gains across different models and 16 dataset variants.
>
> We will add clarifications and further discussion of these points in the revision.

---

> ### Author Response · Authors · 2025-11-21
> **Author response (2/3)**
>
> >**Weakness 4:** The iterative self-training procedure (ten iterations, multiple epochs each) likely increases inference and adaptation time significantly. The paper would benefit from reporting computational overheads, memory costs, and convergence behavior, as well as exploring lightweight or one-shot alternatives to make TTM practical for real-world applications.
>
> **Response:** Thank you for your suggestion. The runtime of TTM scales as $O(T \cdot C_{ft})$, where $T$ is the number of iterations and $C_{ft}$ denotes the model finetuning cost (see detailed runtime analysis below). The memory cost is dominated by the memory footprint of model finetuning. In practice, TTM converges within only a few iterations (e.g., see the right plot in Fig. 4), and already achieves strong performance even with very small $T$ (e.g., 3 or 5; see results below). As a result, the overall computational and memory overhead is **comparable to standard test-time training methods** in the literature (e.g., Sun et al., 2020; Akyürek et al., 2024).
>
> The overhead can be further reduced by adopting efficient finetuning techniques (e.g., PEFT). We leave this as future work, as it is orthogonal to the main contributions of the paper—namely, fixing evaluation metrics and improving model performance.
>
> **Additional results.** While we use $T=10$ in our main experiments, we also evaluate TTM with $T=3$ and $T=5$. As shown below, TTM continues to yield substantial gains even with only a few iterations.
>
> **TTM achieves strong performance even with $T=3$ and $5$ iterations.**
> | **Datasets (Iterations)**       | **GroupScore** | **SimpleMatch** | **TTM**              | **Δ over SimpleMatch (Relative)** | **Error Reduction** |
> |:--------------|:--------------:|:---------------:|:---------------------:|:---------------------------------:|:--------------:|
> | **Winoground ($T=10$)** | 10.25 | 67.00 | **72.50** ± 0.64 | **+5.5 (8.2% ↑)** | **16.7% ↓** |
> | **Winoground ($T=5$)**  | 10.25 | 67.00 | **71.88** ± 1.48 | **+4.9 (7.3% ↑)** | **14.8% ↓** |
> | **Winoground ($T=3$)**  | 10.25 | 67.00 | **71.5** ± 1.19 | **+4.5 (6.7% ↑)** | **13.6% ↓** |
> | **MMVP-VLM ($T=10$)**   | 22.96 | 81.48 | **89.44** ± 0.96 | **+8.0 (9.8% ↑)** | **43.0% ↓** |
> | **MMVP-VLM ($T=5$)**    | 22.96 | 81.48 | **87.04** ± 1.11 | **+5.6 (6.8% ↑)** | **30.0% ↓** |
> | **MMVP-VLM ($T=3$)**    | 22.96 | 81.48 | **85.19** ± 0.91 | **+3.7 (4.6% ↑)** | **20.0% ↓** |
> | **ColorSwap ($T=10$)**  | 30.33 | 88.00 | **94.25** ± 0.43 | **+6.3 (7.1% ↑)** | **52.1% ↓** |
> | **ColorSwap ($T=5$)**   | 30.33 | 88.00 | **93.58** ± 0.86 | **+5.6 (6.3% ↑)** | **46.5% ↓** |
> | **ColorSwap ($T=3$)**   | 30.33 | 88.00 | **93.58** ± 1.01 | **+5.6 (6.3% ↑)** | **46.5% ↓** |
>
> **Detailed runtime analysis.** The runtime of TTM scales as $O(T \cdot C_{ft})$, where $T$ is the number of iterations and $C_{ft}$ denotes the model finetuning cost. Although TTM also performs a matching step within each group, this overhead is negligible compared to finetuning. Specifically:
>
> - The per-group matching cost is $P(k) = O(k^3)$ for $k \times k$ groups (via the Hungarian algorithm) or $P(k) = O(k)$ for $1 \times k$ groups. Since compositional benchmarks use very small group sizes ($k=2$ for $k \times k$ groups or $k=4$ for $1 \times k$ groups), we can safely treat $P(k) = O(1)$.
> - Matching over all $n$ groups therefore costs $O(n)$, which is negligible relative to the finetuning cost $C_{ft}$.
>
> Thus, the overall runtime is dominated by finetuning and scales as $O(T \cdot C_{ft})$. In practice, the finetuning cost $C_{ft}$ can be further reduced via efficient finetuning techniques such as PEFT.

---

> > ### Author Response · Authors · 2025-11-21
> > **Author response (3/3)**
> >
> > >**Weakness 5:** The paper would benefit from a more formal characterization of when and why TTM converges, or how pseudo-label noise affects performance over iterations. Connections to existing theories in semi-supervised or test-time training (e.g., Sun et al., 2020; Gandelsman et al., 2022; Akyürek et al., 2024) could strengthen the methodological depth.
> >
> > **Response:** Thank you for your comment. TTM improves model performance until convergence thanks to the Decay schedule used for data-selection thresholds; please also see our response to Weakness #3 for a detailed discussion. The intuition is as follows:
> >
> > - TTM begins with a high threshold, selecting only high-precision pseudo-labels (few false positives). Learning from these high-quality data reliably improves the model.
> > - As the threshold gradually decreases, TTM incorporates more correctly predicted matchings, which continues to improve performance.
> > - However, a lower threshold also introduces more false-positive pseudo-labels, so the marginal benefit of new data decreases over time.
> >
> > TTM converges once the benefit from newly added correct pseudo-labels no longer outweighs the noise introduced by false positives—that is, when the net gain from pseudo-labeling approaches zero. Empirically, we observe that TTM converges after only a few iterations (e.g., see the right plot of Fig. 4).
> >
> > While we agree that developing a formal theory for test-time training is an interesting direction, such theory is largely absent in the current literature except for highly simplified settings, and it is beyond the scope of the current submission. We leave this for future work.

---

### Official Review · Reviewer_QzQa · 2025-10-31

**Soundness:** 3
**Presentation:** 3
**Contribution:** 3
**Rating:** 6
**Confidence:** 3

**Summary:**

This paper argues that multimodal models' poor performance on compositional reasoning benchmarks is partly due to flawed evaluation metrics that underestimate their true capabilities. The authors first introduce a new evaluation metric, the "group matching score" (GroupMatch), which evaluates the best overall matching within a group and uncovers substantial "hidden competence" in existing models. Building on this, they propose Test-Time Matching (TTM), an iterative, supervision-free self-training algorithm that bootstraps model performance by finetuning on high-confidence pseudo-labels generated at test time. This approach achieves new state-of-the-art results, notably enabling GPT-4.1 to surpass estimated human performance on Winoground and SigLIP-B16 with TTM to outperform GPT-4.1 on MMVP-VLM.

**Strengths:**

1. One strength of the paper is its insightful critique of existing evaluation metrics, such as GroupScore, which it convincingly argues systematically underestimate model capabilities in compositional reasoning. The introduction of the "GroupMatch" score provides a simple, intuitive, and more effective alternative that reveals significant "hidden competence" in existing models.

2. The proposed Test-Time Matching (TTM) algorithm is an effective supervision-free method that iteratively bootstraps model performance. This self-training approach, which uses high-confidence pseudo-labels at test time, provides a practical way to unlock the latent abilities identified by the new metric.

3. The paper presents extensive and strong empirical results across 16 dataset variants, achieving new state-of-the-art (SOTA) performance. Notably, the SimpleMatch method allows GPT-4.1 to surpass estimated human performance on Winoground for the first time, and TTM enables a smaller model (SigLIP-B16) to outperform GPT-4.1 on MMVP-VLM.

**Weaknesses:**

1. The TTM algorithm relies on iterative finetuning at test time, which is computationally expensive and time-consuming. Discussion about the trade-off between compute and performance is needed. For example, compare a small model with more iterations, and a large model with fewer iterations.

2. The performance of TTM appears sensitive to the choice of new hyperparameters, particularly the threshold scheduling used for pseudo-label selection. The paper notes that the initial threshold is "dataset- and model-dependent" and requires careful setting, which may make the method difficult to apply robustly to new datasets or models without costly tuning.

3. GPT-4.1 is closed-sourced. It is necessary to conduct experiments on an open-sourced MLLM.

**Questions:**

Please see the weakness

---

> ### Author Response · Authors · 2025-11-21
> **Author response (1/2)**
>
> Thank you for taking the time to review our paper and for recognizing our contributions in fixing evaluation metrics and substantially improving model performance. Below, we provide detailed responses to your remaining concerns. If our clarifications address these points, we would appreciate it if you could consider updating your assessment.
>
> **Responses to Weaknesses:**
>
> >**Weakness 1:** The TTM algorithm relies on iterative finetuning at test time, which is computationally expensive and time-consuming. Discussion about the trade-off between compute and performance is needed. For example, compare a small model with more iterations, and a large model with fewer iterations.
>
> **Response:** Thank you for your suggestion. We first note that the runtime of TTM scales as $O(T \cdot C_{ft})$, where $T$ is the number of iterations and $C_{ft}$ denotes the model finetuning cost (see detailed runtime analysis below). Importantly, TTM achieves strong performance even with very small $T$ (e.g., 3 or 5; see results below), which keeps the runtime **comparable to standard test-time training methods** in the literature (e.g., Sun et al., 2020; Akyürek et al., 2024).
>
> Regarding the compute-performance trade-off, we observe the following:
>
> - More iterations generally help, but with diminishing returns; as shown in Fig. 4 (right), TTM typically converges after only a few iterations.
> - Larger models are more capable, and in our experiments they consistently achieve higher performance than smaller ones (e.g., Table 1).
>
> Given these observations, when compute is limited, using a larger model with fewer TTM iterations is usually more effective than using a smaller model with many iterations. As a quick empirical check, on the ColorSwap dataset, the 0.7B SigLIP-L16 with 5 iterations achieves \~96, which still outperforms the 0.2B SigLIP-B16 with 10 iterations (\~94) — simply because the larger SigLIP-L16 model is more capable.
>
> **Additional results.** While we use $T=10$ in our main experiments, we also evaluate TTM with $T=3$ and $T=5$. As shown below, TTM continues to yield substantial gains even with only a few iterations.
>
> **TTM achieves strong performance even with $T=3$ and $5$ iterations.**
> | **Datasets (Iterations)**       | **GroupScore** | **SimpleMatch** | **TTM**              | **Δ over SimpleMatch (Relative)** | **Error Reduction** |
> |:--------------|:--------------:|:---------------:|:---------------------:|:---------------------------------:|:--------------:|
> | **Winoground ($T=10$)** | 10.25 | 67.00 | **72.50** ± 0.64 | **+5.5 (8.2% ↑)** | **16.7% ↓** |
> | **Winoground ($T=5$)**  | 10.25 | 67.00 | **71.88** ± 1.48 | **+4.9 (7.3% ↑)** | **14.8% ↓** |
> | **Winoground ($T=3$)**  | 10.25 | 67.00 | **71.5** ± 1.19 | **+4.5 (6.7% ↑)** | **13.6% ↓** |
> | **MMVP-VLM ($T=10$)**   | 22.96 | 81.48 | **89.44** ± 0.96 | **+8.0 (9.8% ↑)** | **43.0% ↓** |
> | **MMVP-VLM ($T=5$)**    | 22.96 | 81.48 | **87.04** ± 1.11 | **+5.6 (6.8% ↑)** | **30.0% ↓** |
> | **MMVP-VLM ($T=3$)**    | 22.96 | 81.48 | **85.19** ± 0.91 | **+3.7 (4.6% ↑)** | **20.0% ↓** |
> | **ColorSwap ($T=10$)**  | 30.33 | 88.00 | **94.25** ± 0.43 | **+6.3 (7.1% ↑)** | **52.1% ↓** |
> | **ColorSwap ($T=5$)**   | 30.33 | 88.00 | **93.58** ± 0.86 | **+5.6 (6.3% ↑)** | **46.5% ↓** |
> | **ColorSwap ($T=3$)**   | 30.33 | 88.00 | **93.58** ± 1.01 | **+5.6 (6.3% ↑)** | **46.5% ↓** |
>
> **Detailed runtime analysis.** The runtime of TTM scales as $O(T \cdot C_{ft})$, where $T$ is the number of iterations and $C_{ft}$ denotes the model finetuning cost. Although TTM also performs a matching step within each group, this overhead is negligible compared to finetuning. Specifically:
>
> - The per-group matching cost is $P(k) = O(k^3)$ for $k \times k$ groups (via the Hungarian algorithm) or $P(k) = O(k)$ for $1 \times k$ groups. Since compositional benchmarks use very small group sizes ($k=2$ for $k \times k$ groups or $k=4$ for $1 \times k$ groups), we can safely treat $P(k) = O(1)$.
> - Matching over all $n$ groups therefore costs $O(n)$, which is negligible relative to the finetuning cost $C_{ft}$.
>
> Thus, the overall runtime is dominated by finetuning and scales as $O(T \cdot C_{ft})$. In practice, the finetuning cost $C_{ft}$ can be further reduced via efficient finetuning techniques such as PEFT.

---

> ### Author Response · Authors · 2025-11-21
> **Author response (2/2)**
>
> >**Weakness 2:** The performance of TTM appears sensitive to the choice of new hyperparameters, particularly the threshold scheduling used for pseudo-label selection. The paper notes that the initial threshold is "dataset- and model-dependent" and requires careful setting, which may make the method difficult to apply robustly to new datasets or models without costly tuning.
>
> **Response:** Thank you for your comment. We would like to clarify that the initial threshold is described as dataset- and model-dependent only when aiming for *optimal* performance. In practice, it is easy to set: a simple rule of thumb is to choose the threshold such that roughly 15%–30% of the groups are matched in the first iteration. Fig. 4 (right) shows this simple heuristic reliably improves model performance.
>
> If a small validation set is available, the initial threshold can also be selected based on matching results on the validation set (see Fig. 2, left and middle).
>
> All other hyperparameters in TTM are straightforward: the ending threshold can be safely set to 0, and both linear and cosine decay schedules work well in practice. We will clarify these points in the revision.
>
> >**Weakness 3:** GPT-4.1 is closed-sourced. It is necessary to conduct experiments on an open-sourced MLLM.
>
> **Response:** We directly use GPT-4.1 because it is better at instruction following and produces more reliable similarity scores (via VQAScore), which is used to compute *both* GroupScore and GroupMatch. Nevertheless, we ran additional experiments on an open-source MLLM SmolVLM-Instruct (2.2B). The trends remain the same—GroupMatch is consistently much higher than GroupScore. Given that **correctness under GroupMatch can be directly converted into correctness under GroupScore**, this further verifies that the widely used GroupScore metric is problematic and artificially deflates model performance.
>
> |   **Metrics**              | **Winoground** | **MMVP-VLM** | **ColorSwap** |
> |:----------------|:--------------:|:------------:|:-------------:|
> | **GroupScore**  | 30.25          | 42.96        | 58.67         |
> | **GroupMatch**  | **80.00**      | **82.96**    | **94.33**     |

---

### Meta-Review · Area_Chair_V4S8 · 2026-01-07

**Summary:**

The paper consists of two main components: (i) an analysis of an existing metric, highlighting its limitations and proposing improvements, and (ii) a test-time self-improvement algorithm that bootstraps model performance without requiring external supervision. Overall, the reviewers appreciated the detailed analysis and insightful critique, and acknowledged the strong empirical results of the proposed method. While some issues were initially raised regarding the sensitivity, complexity, and scalability of the algorithm, as well as misunderstandings of the method, the authors provided extensive empirical results during the discussion period that helped clarify these points. From the AC's perspective, the concerns raised by Reviewer 4Q99 (the primary objector) have largely been addressed, and the AC believes this reviewer would likely raise their score if given the opportunity. Taking all factors into account, the AC finds that the strengths outweigh the weaknesses and therefore recommends acceptance. The AC urges the authors to incorporate the reviewers' feedback into the final version.

**Reviewer Concerns:**

See Above.

**Reviewer Scores:**

See Above.

---

### Decision · Program_Chairs · 2026-01-26

Accept (Poster)